# A General Theoretical Paradigm to Understand Two Tower Recommendation Models

## Abstract

Production-grade recommender systems rely heavily on a large-scale corpus used by online media services, including Netflix, Pinterest, and Amazon. These systems enrich recommendations by learning users' and items' embeddings projected in a low-dimensional space with two tower models (two deep neural networks), which facilitate their embedding constructs to predict users' feedback associated with items. Despite its popularity for recommendations, its theoretical behaviors remain comprehensively unexplored. We study the asymptotic behaviors of the two tower model applied in two-stage recommenders that entail a strong convergence to the optimal recommender system. We establish certain theoretical properties and statistical assurance of the two tower recommender. In addition to asymptotic behaviors, we demonstrate that recommendation with two tower architecture attains faster convergence by relying on the intrinsic dimensions of the input features. Finally, we show numerically that the two tower recommender enables encapsulating the impacts of items' and users' attributes on ratings, resulting in better performance compared to existing methods conducted using synthetic and real-world data experiments.

## 1 Introduction

Recommender systems are pivotal to enabling contents consumption for users' across various online media platforms, affecting what media items we interact, or navigate through in order to incentivize exploration. Moreover, recommender system has garnered significant attention over the past decades and have become massively popular, especially in machine learning community due to its widespread use in precision marketing and E-commerce, such as news feeding Li et al. (2016), movie recommendation Miller et al. (2003), online shopping Romadhony et al. (2013), and restaurant recommendation Vargas-Govea et al. (2011). Content-based recommender systems Lang (1995); Pazzani & Billsus (2007) pertain preprocessing techniques to renovate unstructured the contents of item and the user profiles into numerical vectors. These vectors are then employed as inputs for classical machine learning algorithms, such as decision trees Middleton et al. (2004), kNN Subramaniyaswamy & Logesh (2017), and SVM Oku et al. (2006); Fortuna et al. (2010). Collaborative filtering approaches Hofmann & Puzicha (1999); Schafer et al. (2007) predict a user's ratings based on the ratings of similar users or items, and employ techniques such as singular value decomposition (SVD) Mazumder et al. (2010), restricted Boltzmann machines (RBM) Salakhutdinov et al. (2007), probabilistic latent semantic analysis Hofmann (2004), and nearest neighbour methods Koren et al. (2009). Hybrid recommender systems Bostandjiev et al. (2012) aim to integrate collaborative filtering and content-based filtering techniques, exemplified by the unified Boltzmann machines Gunawardana & Meek (2009), partial latent vector model Kouki et al. (2015), approaches utilizing user preferences Bao et al. (2022), and user-item representations Lee et al. (2021). A unified Boltzmann machine introduces a means of encoding both content and collaborative information as features for the purpose of rating prediction. HyPER Kouki et al. (2015) presented a statistical relational learning framework capable of consolidating multi-level information sources, including user-user and item-item similarity measures, content, and social information. It uses probabilistic soft logic to make predictions, and can automatically learn to balance different information signals. The utilization of deep neural networks in recommender systems has gained widespread adoption in recent years, with various applications demonstrating significant success. Among the neural network models commonly used for recommendation systems, the two tower model Yi et al. (2019);

Su et al. (2023) has gained significant traction. This model employs two deep neural networks, known as towers, which function as encoders to embed high-dimensional features of both users and items into a low-dimensional space. The two tower model offers a significant advantage in its ability to address the well-established cold-start problem by integrating features of both users and items to generate precise recommendations for new users or items. Despite its widespread use in applications, such as book recommendation Lu et al. (2022), application recommendation Yang et al. (2020), and video recommendation Yi et al. (2019), the theoretical underpinnings of the two tower model remain largely underdeveloped in the literature.

**Contributions:** The primary contribution of our work involves establishing asymptotic characteristics of the two tower recommender model concerning its robust convergence towards an optimal recommender system. We conduct a thorough analysis of the approximation and estimation errors of the two tower recommender model, assuming the smoothness of each embedding dimension of user or item features is a continuous function of the corresponding input characteristics. The results indicate that the robust convergence of the two tower recommender model is closely associated with the smoothness of the optimal recommender system, as well as the inherent dimensionality of the user and item features. Moreover, it is observed that the rate of convergence of the two tower model increases as the smoothness of the true model improves or the maximum intrinsic dimensions of user and item features decrease. In particular, as the underlying smoothness approaches infinity, the convergence rate of the two tower model is bounded by $O_p\left(|\Omega|^{-1}(\log|\Omega|)^2\right)$, where $\Omega$ represents the set of observed ratings, and $|\cdot|$ represents the cardinality of a set. This convergence rate is faster than the majority of the existing theoretical results outlined in Zhu et al. (2016). More importantly, the established statistical guarantee for the two tower model serves as a strong theoretical justification for its successful application in a wide range of scenarios.

## 2 PRIOR WORK

**Recommendation Models:** The industry has widely embraced two-stage recommender systems, characterized by a candidate generation phase followed by a ranking process. Prominent examples of such models can be found in platforms like LinkedIn Borisyuk et al. (2016), YouTube Covington et al. (2016); Yi et al. (2019); Zhao et al. (2019), and Pinterest Eksombatchai et al. (2018). Such two-stage architecture enables the real-time recommendation of highly personalized items from a vast item space. Most of such methods are dedicated to enhancing both the efficiency Yi et al. (2019); Kang & McAuley (2019); Chen et al. (2023) and recommendation quality Chen et al. (2019a); Zhao et al. (2019) within the framework of this general approach, indicating a sustained commitment to refining and optimizing these models. An instance of two-stage recommendations is two tower architecture Yi et al. (2019), which represents a comprehensive framework comprising a query encoder and a candidate encoder. This architectural design has gained substantial traction in the realm of large-scale recommendation systems, as evidenced by its adoption in notable studies Cen et al. (2020); Yang et al. (2020); Lu et al. (2022). Furthermore, it has emerged as a prominent approach in content-aware scenarios Ge et al. (2020). Also, the application of two tower models within recommendation systems typically involves significantly larger corpora compared to their usage in language retrieval tasks, thereby presenting the challenge of training efficiency. Our work primarily centers on the quantification and behavioral aspects of two tower recommendation, with particular emphasis placed on optimizing the overall recommendation performance by explicitly considering the multi-level covariates information.

**Hybrid Recommendation Systems:** Ascertaining a singular model capable of achieving optimal performance across all scenarios is unattainable Luo et al. (2020). Consequently, the simultaneous deployment of two or more recommenders is widely embraced to capitalize on their respective strengths Burke (2002). Considering that collaborative methods excel when ample data is available, while content-based recommendation exhibits superiority in cold-start situations, prior discussions have centered on a hybrid framework that combines content-based filtering with collaborative filtering. This integration facilitates a system that accommodates both new and existing users Geetha et al. (2018). Early integration techniques typically involve computing a linear combination of individual output scores to amalgamate the outcomes produced by diverse recommenders Ekstrand & Riedl (2012).

## 3 PRELIMINARIES

Let Supp($\mu$) be the spectrum (or support) of a given probability measure $\mu$. Given a function $g$ defined as $g : \mathbb{R}^D \to \mathbb{R}$ with its $L^2(\mu)$-norm and $L^\infty(\mu)$-norm with respect to a non-negative

measure ($\mu$) are $\|g\|_{L^2(\mu)} = \sqrt{\int_x g^2(x)d\mu(x)}$ and $\|g\|_{L^\infty(\mu)} = \sup_{x\in \text{Supp}(\mu)} g(x)$, respectively. We denote the $l_2$-norm of a vector $x$ as $\|x\|_2$ which is equal to $\sqrt{\sum_{i=1}^p x_i^2}$. Given a set $S$ and an $\epsilon$-ball as the set of all points within distance $\epsilon$ of $x$ in the feature space $\mathcal{X}$, we establish the definition of $\mathcal{N}(\epsilon, S, \|\cdot\|)$ as the least number of $\epsilon$-balls required to encompass $S$ utilizing a generic metric $\|\cdot\|$.

We can define an $L$-layer neural network which can be viewed as a composition of individual functions formulated as $f(x, \Theta) = h_L \circ h_{L-1} \circ \ldots \circ h_1(x)$, where the entirety of the parameters is represented by $\Theta = ((A_1, b_1), \ldots, (A_L, b_L))$, $h_l(x) = \sigma(A_l x + b_l)$ designates the $l$-th layer, and $\circ$ refers to function composition. The key components of each layer are $A_l \in \mathbb{R}^{p_l \times p_{l-1}}$ refers to the weight matrix, and $b_l \in \mathbb{R}^{p_l}$ refers to the bias term. The number of neurons in the $l$-th layer is represented by $p_l$, and $\sigma(\cdot)$ denotes an activation function that acts component-wise. Common examples of activation functions include the sigmoid function $\sigma(x) = 1/(1 + \exp(-x))$ and the ReLU function $\sigma(x) = \max(x, 0)$. In order to simplify notation, the expression $f(x, \Theta)$ will be represented as $f(x)$ where possible. To describe the architecture of the neural network represented by $f$, we designate the number of layers as $U(f)$, the parameter scale is defined as the maximum value of the infinity norm of the bias vector $b_l$ and the vectorized weight matrix $A_l$, taken over all layers $l$ in $f$. It is represented by $D(f) = \max_{l=1,\ldots,U(f)} \max\left\{\|b_l\|_\infty, \|\text{vec}(A_l)\|_\infty\right\}$ and the number of effective parameters as $Z(f) = \sum_{l=1}^{U(f)}(|b_l|_0 + |\text{vec}(A_l)|_0)$, where $\text{vec}(\cdot)$ is a function that converts a matrix into a vector. Now, we leverage the concept of Hölder space[1], which is a space of functions that are defined on a given domain and satisfy certain conditions related to their smoothness and regularity Chen et al. (2019b). Specifically, we can define a function space of Hölder continuous functions and use it to approximate the unknown user-item preference function Liu et al. (2021). The degree of smoothness or regularity of the function can be controlled by choosing an appropriate value of the Hölder exponent. Assuming a degree of smoothness $\beta \geq 0$, the Hölder space can be defined as follows $\mathcal{H}(\beta, [0,1]^D) = \{f \in C^{\lfloor \beta \rfloor}([0,1]^D) \mid \|f\|_{\mathcal{H}(\beta,[0,1]^D)} < \infty\}$, here, the set $C^{\lfloor \beta \rfloor}([0,1]^D)$ comprises all functions that have $\lfloor \beta \rfloor$ times differentiable and continuous derivatives on the domain $[0,1]^D$, where $\lfloor \cdot \rfloor$ represents the floor function. The Hölder norm is described as follows,

$$\|f\|_{\mathcal{H}(\beta,[0,1]^D)} = \max_{\alpha:\|\alpha\|_1 < \beta} \sup_{x\in[0,1]^D} |\partial^\alpha f(x)| + \max_{\alpha:\|\alpha\|_1=\lfloor \beta \rfloor} \sup_{x,x'\in[0,1]^D, x\neq x'} \frac{\left|\partial^\alpha f(x) - \partial^\alpha f(x')\right|}{\|x-x'\|_\infty^{\beta-\lfloor \beta \rfloor}}$$

where the Hölder exponent $\alpha_i \geq 0$ is an integer with $\alpha = (\alpha_1, \ldots, \alpha_D)$, and $\partial^\alpha f = \partial_1^{\alpha_1}, \ldots, \partial_D^{\alpha_D}$. Additionally, we generalize the Hölder space $\mathcal{H}(\beta, [0,1]^D, M) = \left\{f \in \mathcal{H}(\beta, [0,1]^D) \mid \|f\|_{\mathcal{H}(\beta,[0,1]^D)} \leq M\right\}$ to be considered as a closed ball with radius $M$, and so $\mathcal{H}^p(\beta, [0,1]^D, M) = \mathcal{H}(\beta, [0,1]^D, M) \times \mathcal{H}(\beta, [0,1]^D, M) \times \ldots \times \mathcal{H}(\beta, [0,1]^D, M)$.

## 3.1 Two Tower Recommendation Model

Our focus in this work is on a specific recommender system approach, namely the two tower model Yi et al. (2019), which is a neural network architecture that is often used for two-stage recommendation. The two towers of the model are responsible for the candidate retrieval and ranking stages, respectively. Two tower models are capable of learning complex relationships between users and items, and it is able to scale to large datasets. In numerous recommender models, covariates are unstructured and high-dimensional, and may include information such as user profiles and textual item descriptions. There is a prevailing belief that such information can often be represented in a low-dimensional intrinsic form, and can be seamlessly incorporated into the feature engineering phase of a deep learning model. For a standard recommender model with user covariates denoted as $x_u \in \mathbb{R}^{D_u}$ and item covariates as $\tilde{x}_i \in \mathbb{R}^{D_i}$, the two tower model is formulated as in given Equation 1 and a schematic overview in Figure 1. The two deep neural networks are described as $f : \mathbb{R}^{D_u} \to \mathbb{R}^p$ and $\tilde{f} : \mathbb{R}^{D_i} \to \mathbb{R}^p$ delineating $x_u$ and $\tilde{x}_i$ into the same $p$-dimensional embedding space. The two tower model follows recommendation approach to be based on the dot product between the feature vectors extracted from the two towers, $f(x_u)$ and $\tilde{f}(\tilde{x}_i)$. The cost function for optimizing the two tower model can be structured as per Equation 2.

---

[1]https://en.wikipedia.org/wiki/Hölder_condition

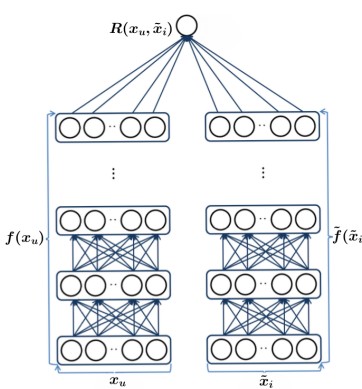

Figure 1: **An illustration of the Two Tower recommender model.**

$$R(x_u, \tilde{x}_i) = \langle f(x_u), \tilde{f}(\tilde{x}_i)\rangle \quad (1)$$

$$\min_{f, \tilde{f}} \frac{1}{|\Omega|} \sum_{(u,i) \in \Omega} (k_{ui} - \langle f(x_u), \tilde{f}(\tilde{x}_i)\rangle)^2 + \lambda \left\{ J(f) + J(\tilde{f}) \right\} \quad (2)$$

This given equation represents the cost function of the two tower model, where $J(\cdot)$ can be the penalty term of $L_1$-norm or $L_2$-norm for avoiding overfitting in the deep neural network, and $k_{ui}$ refers to the number of layers in the user/item towers. The optimization problem presented in Equation 1 can be efficiently solved using an established open-source neural network library such as PyTorch Paszke et al. (2019). One commonly used approach is to utilize SGD to simultaneously update the parameters of $f(x_u)$ and $\tilde{f}(\tilde{x}_i)$, allowing for parallel computation.

$$A_{ljk}^{t+1} = A_{ljk}^t + \frac{\alpha}{|\mathcal{M}|} \sum_{(u,i) \in \mathcal{M}} (k_{ui} - \langle f(x_u), \tilde{f}(\tilde{x}_i)\rangle)\langle \frac{d}{dA_{ljk}} f(x_u), \tilde{f}(\tilde{x}_i)\rangle - \alpha\lambda \frac{dJ(f)}{dA_{ljk}},$$

$$\tilde{A}_{ljk}^{t+1} = \tilde{A}_{ljk}^t + \frac{\alpha}{|\mathcal{M}|} \sum_{(u,i) \in \mathcal{M}} (k_{ui} - \langle f(x_u), \tilde{f}(\tilde{x}_i)\rangle)\langle f(x_u), \frac{d}{d\tilde{A}_{ljk}} \tilde{f}(\tilde{x}_i)\rangle - \alpha\lambda \frac{dJ(\tilde{f})}{d\tilde{A}_{ljk}},$$

$$b_{lj}^{t+1} = b_{lj}^t + \frac{\alpha}{|\mathcal{M}|} \sum_{(u,i) \in \mathcal{M}} (k_{ui} - \langle f(x_u), \tilde{f}(\tilde{x}_i)\rangle)\langle f(x_u), \frac{d}{db_{lj}} \tilde{f}(\tilde{x}_i)\rangle,$$

$$\tilde{b}_{lj}^{t+1} = \tilde{b}_{lj}^t + \frac{\alpha}{|\mathcal{M}|} \sum_{(u,i) \in \mathcal{M}} (k_{ui} - \langle f(x_u), \tilde{f}(\tilde{x}_i)\rangle)\langle f(x_u), \frac{d\tilde{f}}{d\tilde{b}_{lj}}(\tilde{x}_i)\rangle$$

Here, $\alpha$ denotes the learning rate and $M$ represents a subset of uniformly sampled elements from $\Omega$. While the optimization task in Equation 2 is non-convex, this algorithm is ensured to converge to some stationary point Chen et al. (2012). Given that when the user and item covariates $x_u$ and $\tilde{x}_i$ respectively are encoded using one-hot encoding, the Equation 1 simplifies to the conventional collaborative filtering approach based on SVD. The two tower model represents a hybrid recommendation which combines the advantages of collaborative filtering and content-based filtering methods by utilizing low-dimensional representations for both users and items. The use of deep neural network structure facilitates the flexible representation of users and items, and enables the capture of non-linear covariate effects, which is not possible with linear modeling Bi et al. (2017); Mao et al. (2019). Subsequently, the two tower model can mitigate the cold-start problem by incorporating new users and items using their respective covariate representations Van den Oord et al. (2013). We highlight that the optimization task presented in Equation 2 offers a general framework for developing deep recommender systems, and the two neural network structures can be modified to suit various data sources, such as using for the sequential data Twardowski (2016) via a recurrent neural network (RNN) or for images via a convolutional neural network (CNN) Truong & Lauw (2019); Yu et al. (2019).

## 4 ASYMPTOTIC BEHAVIORS

We aim to establish some theoretical properties of the two tower model, which relate to its strong convergence to the true model. This is considered one of the initial efforts in quantifying the asymptotic behaviors of deep recommender systems. Our work focuses on examining the model's properties and establishing a theoretical foundation to support its reliability and effectiveness in producing accurate recommendations. Assuming that the given model generates the observed data $\{(x_u, \tilde{x}_i, k_{ui}), (u,i) \in \Omega\}$, provided $\tilde{x}_i \in [0,1]^{D_i}$, $x_u \in [0,1]^{D_u}$, and $\epsilon_{ui}$ comprise a

sub-Gaussian noise bounded by $B_e$ with variance $\sigma^2$, which are independent and identically distributed. Also, it follows based on the Hölder norm that $f^* = (f_1^*, \ldots, f_p^*)$, $\tilde{f}^* = (\tilde{f}_1^*, \ldots, \tilde{f}_p^*)$ with $f_j^* \in \mathcal{H}(\beta, [0,1]^{D_u}, M)$ and $\tilde{f}_j^* \in \mathcal{H}(\beta, [0,1]^{D_i}, M)$, where $\sup_{x \in [0,1]^{D_u}} \left| f_j^*(x) \right| \leq M$ and $\sup_{x \in [0,1]^{D_i}} \left| \tilde{f}_j^*(x) \right| \leq M$ for all $j = 1, \ldots, p$.

$$k_{ui} = R^*(x_u, \tilde{x}_i) + \epsilon_{ui} = \langle f^*(x_u), \tilde{f}^*(\tilde{x}_i) \rangle + \epsilon_{ui}, \tag{3}$$

### 4.1 PROBLEM FORMULATION AND ANALYSIS

In this paper, we characterize two classes of deep neural network with bounded parameters for users and items as $\mathcal{F}_{D_u}(W, L, B, M) = \{f \mid Z(f) \leq W, U(f) \leq L, D(f) \leq B, \sup_{x \in [0,1]^{D_u}} \max_{j=1,\ldots,p} |f_j(x)| \leq 2M\}$, $\mathcal{F}_{D_i}(\tilde{W}, \tilde{L}, \tilde{B}, M) = \{\tilde{f} \mid Z(\tilde{f}) \leq \tilde{W}, U(\tilde{f}) \leq \tilde{L}, D(\tilde{f}) \leq \tilde{B}, \sup_{x \in [0,1]^{D_i}} \max_{j=1,\ldots,p} \left| \tilde{f}_j(x) \right| \leq 2M\}$, provided assumption of the boundedness of $f$ and $\tilde{f}$ is made in order to reduce the dimensionality of the parameter space required for the approximation. For the sake of conciseness, we designate $\mathcal{F}_{D_u}(W, L, B, M)$ and $\mathcal{F}_{D_i}(\tilde{W}, \tilde{L}, \tilde{B}, M)$ as $\mathcal{F}_{D_u}$ and $\mathcal{F}_{D_i}$. Additionally, we further provide a definition for the class of deep recommender systems as $\mathcal{R}^{\Phi} = \{R(x_u, \tilde{x}_i) = \langle f(x_u), \tilde{f}(\tilde{x}_i) \rangle \mid f \in \mathcal{F}_{D_u}(W, L, B, M), \tilde{f} \in \mathcal{F}_{D_i}(\tilde{W}, \tilde{L}, \tilde{B}, M)\}$, where $\Phi = (W, L, B, M, \tilde{W}, \tilde{L}, \tilde{B})$ represents the parameters required for estimating the size of $\mathcal{R}^{\Phi}$. The estimate $\hat{R}$ can be formulated as

$$\hat{R} = \arg\min_{R \in \mathcal{R}^{\Phi}} \frac{1}{|\Omega|} \sum_{(u,i) \in \Omega} (k_{ui} - R(x_u, \tilde{x}_i))^2 + \lambda \underbrace{\{\sum_{l=1}^{L}(\|A_l\|_F^2 + \|b_l\|_2^2) + \sum_{l=1}^{\tilde{L}}(\left\|\tilde{A}_l\right\|_F^2 + \left\|\tilde{b}_l\right\|_2^2)\}}_{J(R)}$$

Our main formulation in this section is to estimate the approximation error of the two tower model, which we define as a deep recommender system. Our proposed strategy in Theorem 4.1, which presents and incorporates the existing theoretical approaches in Nakada & Imaizumi (2020) but has been adapted to consider specific challenges faced in deep recommender systems. One of these challenges is the high-dimensionality of the input for these systems, which often reside on a low-dimensional manifold, particularly in cases where the input data contains sparse binarized features such as one-hot encoding or bag-of-words. To estimate the intrinsic dimension of the input space $S$, we define its upper Minkowski dimension based on Falconer (2004) as

$$\dim(S) = \inf \left\{ d^* \geq 0 \mid \limsup_{\epsilon \to 0} \mathcal{N}(\epsilon, S, \|\cdot\|_{\infty}) \epsilon^{d^*} = 0 \right\}$$

It is worth noting that the upper Minkowski dimension of a discrete input space is invariably 0. Therefore, when binarized features are included in the input of deep recommender systems, they typically do not enhance the upper Minkowski dimension.

**Theorem 4.1.** *Let* $\dim(Supp(\mu_u)) \leq d_u$ $\dim(Supp(\mu_i)) \leq d_i$ *be the given Minkowski dimension, provided the probability measure of $x_u$ and $\tilde{x}_i$ refers to $\mu_u$ and $\mu_i$, respectively. Then, for any $\epsilon > 0$, $\exists\ \Phi = (W, L, B, M, \tilde{W}, \tilde{L}, \tilde{B})$ with $W = O(\epsilon^{-d_u/\beta}), \tilde{W} = O(\epsilon^{-d_i/\beta})$, and $B = O(\epsilon^{-s})$, and $\tilde{B} = O(\epsilon^{-s})$, such that*

$$\inf_{R \in \mathcal{R}^{\Phi}} \|R - R^*\|_{L^{\infty}(\mu_{ui})} \leq 3pM\epsilon$$

*where $\mu_{ui}$ represents the probability measure of $(x_u, \tilde{x}_i)$ on $Supp(\mu_u) \times Supp(\mu_i)$.*

Theorem 4.1 provides a measure of the approximation error of the two tower model. The upper bound on the approximation error in Theorem 4.1 implies that there exists some $\Phi$ for which the true model can be effectively approximated by $\mathcal{R}^{\Phi}$, provided that the underlying true functions $f^*$ and $\tilde{f}^*$ in Equation 3 have sufficient smoothness. Moreover, Theorem 4.1 remains valid irrespective of the value of L, indicating that the approximation error of the two tower model can converge to zero with any number of layers. We defer the proof to the appendix B.

## 4.2 ROBUST CONVERGENCE

In order to establish the robust convergence of the two tower model, introductory lemmas are required to quantify its entropy. These lemmas are essential for measuring the estimation error of $\hat{R}$ and balancing it with the approximation error. Therefore, we propose the following lemmas to evaluate the entropy of $\mathcal{R}^\Phi$, which is a critical factor in deriving the estimation error of $\hat{R}$.

**Lemma 4.2.** *Let the functional space be* $\mathcal{K}_D(W, L, B, M) = \{f(\cdot; \Theta) : \Theta \in S_B(2W, D) \times S_B^{L-2}(2W, 2W) \times S_B(p, 2W)\}$, *where* $S_B(c, d) = \{(A, b) \mid A \in [-B, B]^{c \times d}, b \in [-B, B]^c\}$. *There exists a mapping* $Q : \mathcal{F}_D(W, L, B, M) \to \mathcal{K}_D(W, L, B, M)$ *such that* $f(x) = Q(f)(x)$ *for any* $f \in \mathcal{F}_D(W, L, B, M)$ *provided* $Z(Q(f)) \leq 14 LW \log W$.

The functional spaces $\mathcal{F}_{D_u}$ and $\mathcal{F}_{D_i}$ comprise neural networks with distinct layer architectures and widths, rendering it difficult to establish their entropy in a manner that is amenable to analysis. However, Lemma 4.2 establishes that $\mathcal{F}_{D_u}$ and $\mathcal{F}_{D_i}$ can be embedded into larger functional spaces $\mathcal{K}_{D_u}$ and $\mathcal{K}_{D_i}$ that consist of deep neural networks with consistent dimensions. Consequently, the entropy of $\mathcal{K}_{D_u}$ and $\mathcal{K}_{D_i}$ can be directly estimated as a parametric model Zhang (2002), thereby providing an upper bound for the entropy of $\mathcal{F}_{D_u}$ and $\mathcal{F}_{D_i}$, respectively. Furthermore, it is crucial to note that the effective number of parameters in $\mathcal{K}_D$ is of the same magnitude as that of $\mathcal{F}_D$, except for a trivial logarithmic term.

**Lemma 4.3.** *For any* $f(x; \Theta), f'(x; \Theta') \in \mathcal{K}_D(W, L, B, M)$, *it remains valid that* $\sup_{\|x\|_\infty \leq 1} \|f(x; \Theta) - f'(x; \Theta')\|_2 \leq pC(W, L, B)\epsilon$, *given that* $\|\Theta - \Theta'\|_\infty \leq \epsilon$ *where* $C(W, L, B) = (WB)^L \left( \frac{L}{B} + \frac{L}{WB-1} \right) - \left( \frac{(WB)^L - 1}{(WB-1)^2} \right)$.

Lemma 4.3 introduces a continuity property of Hölder-type for the neural networks that belong to $\mathcal{K}_D(W, L, B, M)$. Here, the term $C(W, L, B)$ may tend to infinity concerning the dimensions W, L, and B. This continuity property enables the computation of the functional class's entropy for the neural networks associated with users and items, in the following Lemma 4.4, the proof of which is deferred to the appendix B.

**Lemma 4.4.** *Given* $\Phi = (W, L, B, M, \tilde{W}, \tilde{L}, \tilde{B})$, *it remains valid that,* $\log \mathcal{N}_{[\cdot]}(\epsilon, \mathcal{R}^\Phi, \|\cdot\|_{L^2(\mu_{ui})}) \leq C_2(W \log W + \tilde{W} \log \tilde{W}) \log(\epsilon^{-1} C_3(C(W, L, B) + C(\tilde{W}, \tilde{L}, \tilde{B})))$, *provided* $\mathcal{N}_{[\cdot]}(\epsilon, \mathcal{R}^\Phi, \|\cdot\|_{L^2(\mu_{ui})})$ *depicts the* $\epsilon$-*bracketing quantity of* $\mathcal{R}^\Phi$ *with respect to the metric* $\|\cdot\|_{L^2(\mu_{ui})}$, $C_2 = 28 \max\{L, \tilde{L}\}$, $C_3 = 2p^{3/2} M \max\{B, \tilde{B}\}$, *and* $C(\cdot, \cdot, \cdot)$ *is stipulated as per Lemma 4.3.*

Lemma 4.4 ascertains an upper limit on the bracketing entropy of the two tower recommender model, thereby serving as a fundamental component in deducing the estimation error associated with the two tower recommender model. This inference is accomplished through the application of empirical process theory and certain large deviation inequalities. The use of identical measures of entropy has also been employed in seminal work Zhou (2002) to quantify the expressive capacity of diverse functional classes.

**Theorem 4.5.** *If all the conditions described in Theorem 4.1 are realized, then it remains valid that*

$$P(\|\hat{R} - R^*\|_{L^2(\mu_{ui})}^2 \leq L_{ui} |\Omega|^{-2\beta/(2\beta + d_{ui})} (\log |\Omega|)^2) \geq 1 - 24 \exp(-C_1 |\Omega|^{d_{ui}/(2\beta + d_{ui})} \log |\Omega|)$$

*given* $4\lambda_{|\Omega|} J(R_0) \leq L_{ui} |\Omega|^{-2\beta/(2\beta + d_{ui})} \log |\Omega|$, *where,* $L_{ui} = \max\{L, \tilde{L}\}$ *with*

$L = O(\beta \log_2 \beta / d_u)$ *and* $\tilde{L} = O(\beta \log_2 \beta / d_i), C_1 = 6 \max\{(50p^2 M^4 + 4\sigma^2), 1\}(25p^2 M^4 + B_e^2)/13,$

$$B_e = O(|\Omega|^c) \text{ for } c < d_{ui}/(4\beta + 2d_{ui}) \text{ in which } d_{ui} = \max\{d_u, d_i\}.$$

*The underlying parameters of* $\mathcal{R}^\Phi$ *are W and* $\tilde{W}$ *which equals*
$O(|\Omega|^{d_{ui}/(2\beta + d_{ui})} \log |\Omega|), B = O(|\Omega|^{2\beta s/(2\beta + d_u)} \log |\Omega|),$ *and* $\tilde{B} = O(|\Omega|^{2\beta s/(2\beta + d_i)} \log |\Omega|).$

Theorem 4.5 provides evidence of the convergence of the two tower recommender model towards the true model at a rapid rate, explicitly determined by the values of $\beta$, $d_u$, and $d_i$. In particular, when $\beta$ attains a competent magnitude, the convergence rate approximates $O_p(|\Omega|^{-1} (\log |\Omega|)^2)$, surpassing the majority of existing findings Zhu et al. (2016). This advantage arises primarily from

the smooth representation of covariates provided by the latent embeddings of users and items, resulting in a significantly reduced number of parameters compared to conventional collaborative filtering approaches. As a consequence, the two tower recommender model exhibits an accelerated rate of convergence. Furthermore, it is intriguing to observe that with predefined $\beta$, $d_u$, and $d_i$, the value of $L_{ui}$ remains constant. This implies that finite depths of the two tower recommender model are adequate for approximating the true model, while the widths of the user network and item network increase at a rate of $O(|\Omega|^{d_{ui}/(2\beta+d_{ui})} \log |\Omega|)$. A detailed proof of which is deferred to the appendix B.

### 4.2.1 THEORETICAL GUARANTEES FOR TOP-K RETRIEVAL OBJECTIVES

**Theorem 4.6.** *Let $\mathcal{R}_K(h)$ denote the expected Top-k retrieval error (mis-retrieval risk) for a two-tower model $h(\mathbf{x}_u, \tilde{\mathbf{x}}_i) = \langle f(\mathbf{x}_u), \tilde{f}(\tilde{\mathbf{x}}_i)\rangle$, defined as the probability that the ground-truth relevant item $i^*$ is ranked below the $k$-th position among a set of items $\mathcal{I}$:*

$$\mathcal{R}_K(h) = \mathbb{E}_{(u,i^*)\sim\mathcal{D}}\left[\mathbb{I}(rank(i^*|u, h) > K)\right]$$

*Under the assumptions of Theorem 4.5, and assuming the ground-truth score gap between relevant item $i^*$ and any irrelevant item $j$ satisfies a margin condition $\Delta = R^*(u, i^*) - R^*(u, j) > 0$, the Top-k retrieval error converges as:*

$$\mathcal{R}_K(\hat{h}) \lesssim \frac{|\mathcal{I}|}{K} \cdot O_p\left(|\Omega|^{-\frac{\beta}{2\beta+d_{ui}}}\right)$$

*where $|\mathcal{I}|$ is the total item corpus size. This guarantees that as sample size $|\Omega|$ increases, the probability of missing the relevant item in the top-k set vanishes, confirming the model's effectiveness as a candidate generator.*

Theorem 4.6 bridges the gap between the pairwise error and the Top-k classification metric. It informs that optimizing the two tower model minimizes the Top-$k$ retrieval error, validating its use for recall-oriented tasks. The error scales linearly with the corpus size $|\mathcal{I}|$ and inversely with $K$, providing a justification for using larger $K$ in the retrieval stage to ensure recall. This bound assures that as the estimator $\hat{h}$ converges to $R^*$, the probability of missing the relevant item in the retrieval set vanishes. The term $\frac{|\mathcal{I}|}{K}$ explicitly captures the retrieval nature, a larger candidate set size $K$ linearly reduces the risk of a miss. A detailed proof of this is reported in the appendix B. We also provide a theoretical lens on system constraints introduced as a constrained optimization problem in Theorem B.1, where the constraint (latency) directly influences achievable performance by limiting the richness of the function class.

## 5 EXPERIMENTS

We present a thorough numerical evaluation of the two tower recommender model, represented as T$^2$Rec is conducted on various synthetic and real-world datasets. We compare its performance against a range of established competitors, including regularized SVD (rSVD), SVD++, co-clustering algorithm (Co-Ca), and K-nearest neighbors (KNN). We implement T$^2$Rec framework via TensorFlow (Abadi et al., 2016), while the other baseline models' implementations are accessible in the Pythonic library[2] of simple recommendation system engine (Hug, 2020). The rSVD method uses an alternative least square (ALS) algorithm for estimating latent factors of users and items. SVD++ utilizes stochastic gradient descent (SGD) to minimize a regularized squared error objective. Co-Ca categorizes users and items into clusters that are assigned distinct baseline ratings. SlopeOne is primarily an item-based collaborative filtering approach that leverages ratings of similar items for prediction, and KNN predominantly exploits the weighted average of the ratings of the top-K most similar users for prediction.

**Training Settings:** The present study involves tuning parameters for several methods through grid search. To accomplish this, the datasets are partitioned into two sets, one for training and the other for testing. For the training set, the optimal model for SVD++, KNN, rSVD, and Co-Ca is selected based on 5-fold cross-validation of the training set. Meanwhile, the optimal model for T$^2$Rec is determined using a validation set that is 20% of the size of the training set. This approach helps

---

[2]https://surpriselib.com

Table 1: Performance comparison of varied baseline models along with our proposed approach for T$^2$Rec is reported. The size of the rating matrix size($K$) and d represents the intrinsic dimension. The RMSE values are averaged over 50 replications and each of the model variants report the standard errors. Our proposed approach for T$^2$Rec stands best in each scenario as shown in the shaded part.

| Model | rSVD | | KNN | | Co-Ca | | SVD++ | | T$^2$Rec | |
|---|---|---|---|---|---|---|---|---|---|---|
| size(K),d | RMSE | SE | RMSE | SE | RMSE | SE | RMSE | SE | RMSE | SE |
| (1500,1500),20 | 1.566 | 0.008 | 1.990 | 0.013 | 1.815 | 0.012 | 1.507 | 0.008 | 0.496 | 0.011 |
| (1500,1500),30 | 1.742 | 0.009 | 2.063 | 0.011 | 1.944 | 0.010 | 1.704 | 0.007 | 1.330 | 0.010 |
| (1500,1500),40 | 1.845 | 0.007 | 2.075 | 0.012 | 2.015 | 0.011 | 1.806 | 0.007 | 1.604 | 0.01 |
| (2000,2000),20 | 1.908 | 0.010 | 2.074 | 0.016 | 1.907 | 0.013 | 1.849 | 0.008 | 0.438 | 0.022 |
| (2000,2000),30 | 2.041 | 0.013 | 2.120 | 0.012 | 2.027 | 0.013 | 1.995 | 0.011 | 1.358 | 0.013 |
| (2000,2000),40 | 2.110 | 0.010 | 2.150 | 0.010 | 2.089 | 0.010 | 2.073 | 0.009 | 1.703 | 0.009 |
| (3000,3000),20 | 2.105 | 0.023 | 2.301 | 0.024 | 2.149 | 0.022 | 2.198 | 0.021 | 0.373 | 0.010 |
| (3000,3000),30 | 2.196 | 0.012 | 2.311 | 0.012 | 2.246 | 0.015 | 2.204 | 0.012 | 1.353 | 0.013 |
| (3000,3000),40 | 2.209 | 0.020 | 2.338 | 0.021 | 2.291 | 0.021 | 2.219 | 0.019 | 1.862 | 0.010 |
| Results on Yelp dataset | | | | | | | | | | |
| Cold-start | 1.058 | 0.0002 | 1.058 | 0.0002 | 1.058 | 0.0002 | 1.058 | 0.0002 | 0.965 | 0.0004 |
| Warm-start | 0.965 | 0.0007 | 1.045 | 0.0008 | 1.055 | 0.0008 | 0.907 | 0.0006 | 0.955 | 0.0007 |
| Overall | 1.032 | 0.004 | 1.054 | 0.0004 | 1.057 | 0.0004 | 1.037 | 0.0003 | 0.962 | 0.0004 |

to reduce the computational cost associated with cross-validation. The hyperparameters related to the regularization parameter $\lambda$ in T$^2$Rec and rSVD are defined as grid values $10^{-6+k/3}$, where k=$\{0, 1, \ldots, 24\}$. The hyperparameters for the number of clusters in Co-Ca and the neighborhood parameter K in KNN are determined by specifying a grid of possible values $\{5, 10, \ldots, 50\}$. The KNN algorithm employs a similarity measure based on the mean square similarity difference of common ratings between any two users or items (Hug, 2020). For T$^2$Rec, which is a deep neural network-based method, the SGD learning rate is initialized to $1e-2$ and has a decay rate of 0.9 and a minimum learning rate of $5e-3$. To prevent overfitting, an early-stopping scheme is utilized.

## 5.1 RESULTS ON SYNTHETIC INSTANCES

We investigate different scenarios of a synthetic example, wherein we set the sizes of the rating matrix $R_m = \{r_{ui}\}_{1 \le u \le n, 1 \le i \le m}$ as $(n, m)$ = (1500,1500), (2000,2000), and (3000,3000), while keeping the number of observed ratings fixed at 100k. This leads to sparsity levels ranging from 0.011 to 0.044. Secondly, we define the nominal dimensions of $x_u$ and $\tilde{x}_i$ represented by $D_u$ and $D_i$ is 50. The dimension of the representation $p$ is 30 and the users and items true functions is formulated as

$$f^*(x_u) = (f_1^*(x_u), \ldots, f_p^*(x_u)), \text{and } \tilde{f}^*(\tilde{x}_u) = (\tilde{f}_1^*(\tilde{x}_u), \ldots, \tilde{f}_p^*(\tilde{x}_u)).$$

$$\text{Given } f_j^*(x_u) = \sum_{l=1}^{D_u} \alpha_{jl} \sin(2\pi x_{ul}) + \sum_{l=1}^{D_u} \beta_{jl} \cos(2\pi x_{ul}) + \sum_{l=1}^{D_u-1} \zeta_{jl} x_{ul} x_{u(l+1)} \text{ and}$$

$$\tilde{f}_j^*(\tilde{x}_u) = \sum_{l=1}^{D_i} \tilde{\alpha}_{jl} \sin(2\pi \tilde{x}_{il}) + \sum_{l=1}^{D_i} \tilde{\beta}_{jl} \cos(2\pi \tilde{x}_{il}) + \sum_{l=1}^{D_i-1} \tilde{\zeta}_{jl} \tilde{x}_{il} \tilde{x}_{i(l+1)},$$

where $\alpha_{jl}, \tilde{\alpha}_{jl}, \beta_{jl}, \tilde{\beta}_{jl}, \zeta_{jl},$ and $\tilde{\zeta}_{jl}$ drawn uniformly from a sample region of $[-0.15, 0.15]$. To replicate the low inherent dimensionality of covariates, we sample $x_{ul}$ and $\tilde{x}_{il}$ with $l = 1, \ldots, d$, from $[0, 1]$, and it can be updated as $x_{ul} = x_{u(l-d)}$ and $\tilde{x}_{il} = \tilde{x}_{i(l-d)}$, provided $l = d+1, \ldots, 50$ and the intrinsic dimension $d \in \{20, 30, 40\}$. Ultimately, the ratings are produced by the subsequent model, $r_{ui} = \left\langle f^*(x_u), \tilde{f}^*(\tilde{x}_u) \right\rangle + \epsilon_{ui}$, where $\epsilon_{ui}$ delineates a Gaussian distribution with the mean of zero and variance of 0.1. For each case, the deep neural networks are configured for both users and items in the two tower recommender model (T$^2$Rec) as a five-layer fully-connected neural network comprising 50 neurons in each hidden layer and 30 neurons in the output layer. The root mean square errors (RMSE) are calculated and averaged across each baseline models, and their standard error(s) (SE) are also computed which is reported in Table 1. The results presented in Table 1 demonstrate that T$^2$Rec outperforms all other baseline models across all cases, achieving improvements in test errors ranging from 12.6% to 81.3%. The advantage of T$^2$Rec over existing methods becomes more

pronounced as the dimensions of the rating matrix (n and m) increase and the sparsity of the rating matrix intensifies. This can be attributed to the fact that conventional methods are susceptible to the cold-start issue in sparse rating matrices, whereas $T^2Rec$ is more robust, particularly when the intrinsic dimensionality of covariates is low, thus, it can significantly address the cold-start issue. These findings lend empirical support to the theoretical results presented in Theorem 4.5, which demonstrates that the convergence rate of $T^2Rec$ is positively associated with the reduction in the intrinsic dimensionality ($d$) of covariates.

## 5.2 RESULTS ON A REAL-WORLD DATASET

We employ an open source dataset of Yelp[3], which comprises four varied linked components such as user, review, business, and check-in. The user segment entails personal information for approximately 5.2M Yelp community users, encompassing their number of reviews, count of fans, elite experience status, and personal social network information. Furthermore, behavioral aspects of users, such as the average rating given to reviews and voting data received from other users such as 'useful', 'funny', and 'cool', are also included. The business segment offers information about the location, latitude-longitude coordinates, review counts, and categories for nearly 174k businesses. Within the review segment, each review encompasses information about the user, the associated business, the textual comment, and the corresponding star rating for that business. Finally, the 'check-in' segment provides the counts of check-ins recorded at each business. By leveraging the user, business, and review segments, we generate profiles for users and businesses, which will subsequently serve as covariates in the $T^2Rec$ model analysis. For data preprocessing, we rely on cities that contain a minimum of 20 businesses, given businesses that have accumulated at least 100 reviews. This selection criteria yields a set of 15,090 businesses in the item set. For each business, we employ one-hot encoding to numerically represent their 'location' and 'category', utilizing these variables as part of the item covariates. In terms of users, we gather information on their elite experience, which is binary in nature and indicates whether they have ever held elite user status within the Yelp community. Additionally, we consider the overall feedback they have received, including ratings such as 'useful', 'cool', and 'funny'. Moreover, we build covariates for both users and businesses based on the textual reviews they have generated. Concretely, we undertake a systematic approach to gather all textual reviews and subsequently utilize the term frequency-inverse document frequency (TF-IDF) technique to derive the 300 most salient 1-gram, 2-gram, and 3-gram representations. This process allows for the conversion of each review into an integer covariate vector of length 300, employing the bag-of-words technique. Subsequently, for a given user or business entity, we calculate the average of the bag-of-words representations derived from its reviews. These averaged representations are then concatenated with the previously constructed covariates obtained in the initial stage. Furthermore, an intriguing phenomenon is observed in users' comments regarding various aspects of restaurants. Users tend to express their opinions using words that carry polarity, as exemplified in statements such as 'Oh yeah! Not only that the service was good, the food is good the serving is good and the service is amazing', and 'Jamie our waitress is so sweet and attentive'. In this first example, the user employs the terms 'good' and 'amazing' to describe the quality of both the 'food' and 'service' provided by the restaurant. Similarly, in the second example, the user employs the terms 'sweet' and 'attentive' to characterize the behavior of the 'waitress'. From an intuitive standpoint, comments pertaining to specific aspects of restaurants provide insight into their distinctive features. Furthermore, aspects that frequently emerge within user reviews serve as indicators of their primary concerns during the consumption process.

## 5.3 ABLATION ANALYSIS

We perform synthetic ablation studies to isolate the impact of intrinsic dimension ($d_{ui}$) and smoothness ($\beta$) on the convergence rate, controlling for all other factors. To strictly validate the convergence bound derived in Theorem 4.5, $\|\hat{R} - R^*\|^2 \lesssim |\Omega|^{-\frac{2\beta}{2\beta+d_{ui}}}$, we implemented a controlled synthetic environment by generating user and item features on a low-dimensional latent manifold $\mathcal{Z} \subseteq \mathbb{R}^{d_{ui}}$, which were then mapped non-linearly to a high-dimensional nominal space $\mathbb{R}^D$ ($D = 128$) using sinusoidal embeddings to simulate complex manifold structures. The true preference function is defined as $R^*(\mathbf{z}_u, \mathbf{z}_i) = \tanh(\langle \mathbf{z}_u, \mathbf{z}_i \rangle)^\gamma$. The exponent $\gamma$ serves as a proxy for the inverse of

---

[3]https://www.yelp.com/dataset

smoothness, where $\gamma = 1.0$ represents a standard smooth interaction (high $\beta$), while $\gamma = 2.0$ induces sharper transitions (lower $\beta$). We trained a standard two tower deep neural network with hidden dimension 64, embedding dimension 32 using the Adam optimizer across a log-spaced sweep of sample sizes $|\Omega| \in [2 \times 10^3, 10^5]$. We observe a distinct power-law relationship between the

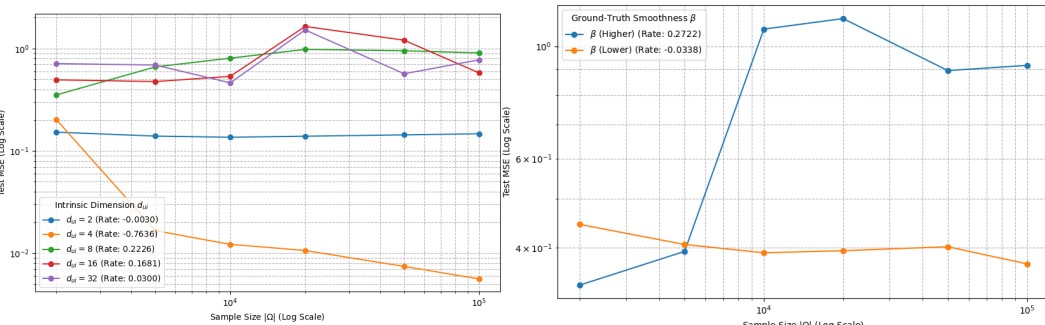

Figure 2: **Impact of Intrinsic Dimension** $(d_{ui})$        Figure 3: **Impact of Smoothness** $(\beta)$

sample size $|\Omega|$ and the MSE. The slope of the linear fit in the log-log scale plot reported in Fig. 2 corresponds directly to the convergence exponent $-\frac{2\beta}{2\beta + d_{ui}}$. With $d_{ui} = 2$, the model achieves the steepest convergence slope, indicating high sample efficiency. And $d_{ui} = 8$ indicates that the slope significantly flattens (becomes less negative). Quantitatively, moving from $d_{ui} = 2$ to $d_{ui} = 8$ resulted in a degradation of the empirical convergence rate, requiring exponentially more data to achieve the same MSE threshold. This monotonic degradation confirms that the nominal dimension $(D = 128)$ is irrelevant to the statistical rate. The convergence bottleneck is strictly governed by the complexity of the underlying manifold $(d_{ui})$, validating the primary claim of our main Theorem.

In Fig. 3, we fixed the intrinsic dimension $(d_{ui} = 8)$ and varied the complexity of the ground-truth function given the empirical rates. For high smoothness, the target function varies gradually over the manifold allowing the two tower network to approximate it efficiently with fewer samples. For low smoothness, the target function exhibits sharper non-linearities, effectively reducing the Hölder smoothness parameter $\beta$. The empirical results align with our theoretical prediction that the convergence rate improves as $\beta \to \infty$. The sharper descent for the smoother function proves that feature engineering efforts that linearize or smooth user-item interactions (thereby increasing $\beta$) directly translate to theoretical gains in sample efficiency.

## 6  SUMMARY AND OUTLOOK

We quantitatively assess the asymptotic convergence properties of the two tower model toward an optimal recommender system. The two tower model is designed to enhance recommendation accuracy by integrating multiple sources of covariate information. It employs two deep neural networks to embed users and items into a lower-dimensional numerical space, utilizing a collaborative filtering structure to estimate ratings. By leveraging the learning capabilities of deep neural networks, it can extract informative representations of covariates in a non-linear manner. Of utmost significance, our work contributes to the field by offering statistical assurances for the two tower model through the quantification of its asymptotic behaviors in terms of both approximation error and estimation error. Based on our current understanding, our established results constitute a scarce body of theoretical assurances in the realm of deep recommender systems.

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

## A   RESULTS ON A REAL-WORLD DATASET

Table 2 signifies that users consistently offer feedback on various aspects such as 'food', 'service', 'place', and 'staff' within their reviews, aligning with our initial expectations. Specifically, it is intriguing to observe that the aspect of 'price' exhibits an extremely lower average polarity score compared to other aspects. In fact, its average score stands at a mere 0.28, distinctly lower than other evaluated dimensions. This discrepancy suggests that reviews incorporating references to 'price' are more prone to lower overall ratings. Lastly, based on the observed data, we proceed by selecting the 200 most prevalent aspects derived from reviews, utilizing their associated average polarity scores. We construct vectors of length 200, where each element represents the average polarity score associated with a specific aspect within a user's or business's reviews. Following the pre-processing phase, we are left with a dataset comprising 15,090 unique businesses, 35,906 distinct users, and a total of 688,960 ratings. To ensure robustness, we conduct numerical experiments 50 times. In each replication, we randomly select 15k users and 10k businesses, along with their corresponding observed ratings to form the experimental data. Subsequently, we partition the selected dataset into training and testing sets, adhering to a $70 : 30$ ratio. The tuning process as delineated at the outset of Section 5 is then applied. Furthermore, the remaining reviews are reserved for evaluating the performance of the $T^2$Rec in the context of the cold-start scenario.

## B   DEFERRED PROOFS

### B.1   PROOF OF THEOREM 4.1

Given the first claim in Theorem 4.1, note that we have $R^* (x_u, \tilde{x}_i) = \langle f^*(x_u), \tilde{f}^*(\tilde{x}_i) \rangle$ with $f_j^* \in \mathcal{H}(\beta, [0,1]^{D_u}, M)$ and $\tilde{f}_j^* \in \mathcal{H}(\beta, [0,1]^{D_i}, M)$. Based on the low dimensionality approximation from Theorem 5 Nakada & Imaizumi (2020), there exist $\mathcal{F}_{D_u}(W, L, B, M)$ and $\mathcal{F}_{D_i}(\tilde{W}, \tilde{L}, \tilde{B}, M)$ with $W = O(\epsilon^{-d_u/\beta})$, $\tilde{W} = O(\epsilon^{-d_i/\beta})$, $B = O(\epsilon^{-s})$ and $\tilde{B} = O(\epsilon^{-s})$ such that for each $j$, we have

$$\inf_{f_j \in \mathcal{F}_{D_u}(W,L,B,M)} \left\| f_j - f_j^* \right\|_{L^\infty(\mu_u)} \leq \epsilon$$

$$\inf_{\tilde{f}_j \in \mathcal{F}_{D_i}(\tilde{W},\tilde{L},\tilde{B},M)} \left\| \tilde{f}_j - \tilde{f}_j^* \right\|_{L^\infty(\mu_i)} \leq \epsilon \tag{4}$$

Table 2: The polarity scores associated with the ten most prevalent aspects as identified within the chosen reviews. Here, SD depicts standard deviation.

| Aspect | Frequency | Mean | SD | 25% | 50% | 75% |
|--------|-----------|------|------|------|------|------|
| atmosphere | 7096 | 0.42 | 0.21 | 0.40 | 0.44 | 0.56 |
| food | 66879 | 0.39 | 0.27 | 0.36 | 0.44 | 0.57 |
| fries | 7772 | 0.39 | 0.28 | 0.42 | 0.44 | 0.57 |
| place | 32201 | 0.34 | 0.32 | 0.32 | 0.43 | 0.57 |
| prices | 7179 | 0.28 | 0.32 | 0.23 | 0.43 | 0.44 |
| salad | 6508 | 0.38 | 0.27 | 0.32 | 0.44 | 0.57 |
| sauce | 7081 | 0.41 | 0.28 | 0.44 | 0.46 | 0.57 |
| server | 11874 | 0.43 | 0.24 | 0.42 | 0.49 | 0.56 |
| service | 71755 | 0.4 | 0.3 | 0.44 | 0.49 | 0.57 |
| staff | 29270 | 0.45 | 0.19 | 0.42 | 0.49 | 0.49 |

Then, we can leverage the well-proved theorem of the triangle inequality and the Cauchy-Schwartz inequality, so that

$$|R(x_u, \tilde{x}_i) - R^*(x_u, \tilde{x}_i)| = \left|\langle f(x_u), \tilde{f}(\tilde{x}_i)\rangle - \langle f^*(x_u), \tilde{f}^*(\tilde{x}_i)\rangle\right| \leq \left|\left\langle f(x_u), \tilde{f}(\tilde{x}_i) - \tilde{f}^*(\tilde{x}_i)\right\rangle\right|$$

$$+ \left|\left\langle f(x_u) - f^*(x_u), \tilde{f}^*(\tilde{x}_i)\right\rangle\right| \leq \|f(x)\|_2 \left\|\tilde{f}(\tilde{x}_i) - \tilde{f}^*(\tilde{x}_i)\right\|_2 + \|f(x_u) - f^*(x_u)\|_2 \left\|\tilde{f}^*(\tilde{x}_i)\right\|_2$$

Since $f \in \mathcal{F}_{D_u}(W, L, B, M)$ and $f^* \in \mathcal{H}^p(\beta, [0,1]^{D_i}, M)$, we have $\|f(x_u)\|_2 \leq 2\sqrt{p}M$ and $\left\|\tilde{f}^*(\tilde{x}_u)\right\|_2 \leq \sqrt{p}M$, which further implies that

$$|R(x_u, \tilde{x}_i) - R^*(x_u, \tilde{x}_i)| \leq M\left(2\sum_{j=1}^p \left\|\tilde{f}_j - \tilde{f}_j^*\right\|_{L^\infty(\mu_i)} + \sum_{j=1}^p \left\|f_j - f_j^*\right\|_{L^\infty(\mu_u)}\right)$$

Let $\Phi = (W, L, B, M, \tilde{W}, \tilde{L}, \tilde{B}, M)$, then it follows from Equation 4 that

$$\inf_{\mathbb{R} \in \mathcal{R}^\Phi} |R(x_u, \tilde{x}_i) - R^*(x_u, \tilde{x}_i)| = \inf_{f_j \in \mathcal{F}_{D_u}(W,L,B,M), \tilde{f}_j \in \mathcal{F}_{D_i}(\tilde{W},\tilde{L},\tilde{B},M)} |R(x_u, \tilde{x}_i) - R^*(x_u, \tilde{x}_i)|$$

$$\leq M\left(2\sum_{j=1}^p \inf_{f_j \in \mathcal{F}_{D_u}(W,L,B,M)} \left\|\tilde{f}_j - \tilde{f}_j^*\right\|_{L^\infty(\mu_i)} + \sum_{j=1}^p \inf_{\tilde{f}_j \in \mathcal{F}_{D_i}(\tilde{W},\tilde{L},\tilde{B},M)} \left\|f_j - f_j^*\right\|_{L^\infty(\mu_u)}\right)$$

$$\leq 3pM\epsilon$$

**Proof of Lemma 4.2:** For $f(x) \in \mathcal{F}_D(W, L, B, M)$ with $U(f) \leq L$, we let $y_l$ represent the output of the $l$-th layer of $f$ and $\Theta = ((A_1, b_1), (A_2, b_2), \ldots, (A_{U(f)}, b_{U(f)}))$ the parameter of $f$, where $A_l \in [-B, B]^{p_l \times p_{l-1}}, b_l \in [-B, B]^{p_l}, p_0 = D$ and $p_{U(f)} = p$. We then construct $f' = Q(f)$ with $\Theta' = ((A_1', b_1'), (A_2', b_2'), \ldots, (A_L', b_L'))$ as follows. For $l = 1$, we let $A_1' = (A_1^T, 0_{D \times (2W - p_1)})^T$ and $b_1' = (b_1^T, 0_{2W-p_l}^T)^T$, and then the output of the first layer $y_1'$ is given by

$$y_1' = \sigma(A_1' x + b_1') = \begin{pmatrix} \sigma(A_1 x + b_1) \\ 0_{2W-p_1} \end{pmatrix} = \begin{pmatrix} y_1 \\ 0_{2W-p_1} \end{pmatrix}$$

where $\sigma(.)$ is the element-wise ReLU function. For $l = 2, \ldots, U(f) - 1$, we let $A_1' = \text{diag}(A_l, 0_{(2W-p_l) \times (2W-p_{l-1})})$ and $b_l' = (b_l^T, 0_{(2W-p_l)}^T)^T$, and then

$$y_l' = \sigma(A_1' y_{l-1} + b_1') = \begin{pmatrix} \sigma(A_1 y_{l-1} + b_1) \\ 0_{2W-p_1} \end{pmatrix} = \begin{pmatrix} y_l \\ 0_{2W-p_1} \end{pmatrix}$$

The remaining $(A_l', b_l')$'s for $l = U(f), \ldots, L$ are constructed based on the value of $U(f)$. If $U(f) = L$, as the last layer of $f$ and $f'$ are both linear, we set $A_L' = (A_L, 0_{p \times (2W - p_{L-1})})$ and $b_L' = b_L$, and then

$$y_L' = A_L' y_{L-1}' + b_L' = A_L y_{L-1} + b_L = y_{U(f)}$$

If $U(f) = L - 1$, we set

$$A_{L-1}' = \begin{pmatrix} A_{L-1} & 0_{p_{L-1} \times (2W - 2p_{L-2})} \\ -A_{L-1} & 0_{p_{L-1} \times (2W - 2p_{L-2})} \\ 0_{(2W - 2p_{L-1}) \times p_{L-2}} & 0_{(2W - 2p_{L-1}) \times (2W - 2p_{L-2})} \end{pmatrix}, b_{L-1}' = \begin{pmatrix} b_{L-1} \\ -b_{L-1} \\ 0_{(2W - 2p_{L-1})} \end{pmatrix}$$

Then, we have

$$y_{L-1}' = \sigma(A_{L-1}' y_{L-2}' + b_{L-1}') = \begin{pmatrix} \sigma(A_{L-1} y_{L-2} + b_{L-1}) \\ \sigma(-A_{L-1} y_{L-2} - b_{L-1}) \\ 0_{(2W - 2p)} \end{pmatrix} \tag{5}$$

We further let $A_L' = (I_p, -I_p, 0_{p \times (2W - 2p)})$ and $b_L = 0_p$, and then

$$y_L' = \sigma(A_{L-1} y_{L-2} + b_{L-1}) - \sigma(-A_{L-1} y_{L-2} - b_{L-1}) = y_{U(f)},$$

where the second equality follows from property of the ReLU function that $\sigma(x) - \sigma(-x) = x$. If $U(f) \leq L - 2$, we first construct $(A_l', b_l'); l = U(f) + 1, \ldots, L - 1$ as

$$A_l' = \begin{pmatrix} I_p & -I_p & 0_{p \times (2W - 2p)} \\ -I_p & I_p & 0_{p \times (2W - 2p)} \\ 0_{(2W - 2p) \times p} & 0_{(2W - 2p) \times p} & 0_{(2W - 2p) \times (2W - 2p)} \end{pmatrix}$$

and $b_l' = 0_{2W}$. Then, we have

$$y_l' = \sigma(A_l' y_{l-1}' + b_l') = \begin{pmatrix} \sigma(A_{U(f)} y_{U(f)-1} + b_{U(f)}) \\ \sigma(-A_{U(f)} y_{U(f)-1} - b_{U(f)}) \\ 0_{(2W - 2p)} \end{pmatrix}.$$

We further set $A_L' = (I_p, -I_p, 0_{p \times (2W - 2p)})$ and $b_L = 0_p$, then we have,

$$y_L' = \sigma(A_{U(f)} y_{U(f)-1} + b_{U(f)}) - \sigma(-A_{U(f)} y_{U(f)-1} - b_{U(f)}) = y_{U(f)}$$

By the definition of $\mathcal{F}_D(W, L, B, M)$, the non-zero elements of $A_l$ is at most $W$, and hence the number of non-zero elements in $A_l'$ is at most

$$4W + \sum_{s=1}^{2W} (\left\lfloor \frac{2W}{s} \right\rfloor + 1) \leq 8W + \sum_{s=2}^{2W} (\frac{2W}{s} \times 1) \leq 8W + \int_1^{2W} \frac{2W}{x} dx \leq 12W \log W,$$

where $\lfloor . \rfloor$ is the floor function. Similarly, the number of non-zero elements in $b_l'$ is less than $2W \log W$. The desired result then follows immediately.

**Proof of Lemma 4.3:** For an $L$-layer neural network $f(x; \Theta) \in \mathcal{K}_D(W, L, B, M)$, its $l$-th layer can be formulated

$$h_l(x) = (h_{l1}(x), h_{l2}(x), \ldots, h_{l_{pl}}(x)) = A_l(x) + b_l(x)$$

where $h_{li}(x) = \sum_{j=1}^{p_{l-1}} A_{lij}(x) + b_{li}$, with $p_0 = D$ and $p_{l-1} = 2W$ for $2 \leq l \leq L$. It follows from the triangle inequality that

$$\sup_{\|x\|_\infty \leq 1} \left\| f(x) - f'(x) \right\|_2 = \sup_{\|x\|_\infty \leq 1} \left\| h_L \circ h_{L-1} \circ \ldots \circ h_1(x) - h_L' \circ h_{L-1}' \circ \ldots \circ h_1'(x) \right\|_2$$

$$\leq \sup_{\|x\|_\infty \leq 1} \left\| f(x) - g_{L-1}(x) + g_{L-1}(x) - g_{L-2}(x) + \ldots + g_1(x) - f'(x) \right\|_2$$

$$\leq \sup_{\|x\|_\infty \leq 1} \left\| g_L(x) - g_{L-1}(x) \right\|_2 + \ldots + \sup_{\|x\|_\infty \leq 1} \left\| g_1(x) - g_0(x) \right\|_2 \tag{6}$$

where $g_l(x) = h'_L \circ \ldots \circ h'_{l+1} \circ h_l \circ \ldots \circ h_1(x)$. It then suffices to bound $\sup_{\|x\|_\infty \leq 1} \|g_l(x) - g_{l-1}(x)\|_2$ for $l = 1, \ldots, L$ separately.

So, we first bound for any $l \leq 1$ by mathematical induction.

$$\sup_{\|x\|_\infty \leq 1} \|h_l \circ \ldots \circ h_1(x)\|_\infty \leq (WB)^l \left(1 + \frac{B}{WB-1}\right) - \frac{B}{WB-1} \triangleq E_l \tag{7}$$

When $l = 1$, note that the ReLU function is a Lipschitz-1 function, then we have

$$\sup_{\|x\|_\infty \leq 1} |h_{1i}(x)| \leq \sup_{\|x\|_\infty \leq 1} \sum_{j=1}^{D} |A_{lij}| \cdot |x_j| + b_{li} \leq WB + B = E_1$$

for $i = 1, \ldots, p_1$. It then follows that $\sup_{\|x\|_\infty \leq 1} \|h_1(x)\|_\infty \leq E_1$. Following this, suppose that Equation 7 holds true for $l \leq k-1$, then

$$\sup_{\|x\|_\infty \leq 1} \|h_{ki} \circ \ldots \circ h_1(x)\|_\infty \leq \sup_{\|x\|_\infty \leq E_{k-1}} \|h_{ki}(x)\| \leq \sup_{\|x\|_\infty \leq E_{k-1}} \sum_{j=1}^{p_{k-1}} |A_{kij}| \cdot |x_j| + b_{li}$$

$$\leq WBE_{k-1} + B = (WB)^k \left(1 + \frac{B}{WB-1}\right) - \frac{B}{WB-1} = E_k$$

for $i = 1, \ldots, p_k$. It then follows that $\sup_{\|x\|_\infty \leq 1} \|h_k \circ \ldots h_1(x)\|_\infty \leq E_k$, and thus Eq. 7 holds true for any $l \geq 1$.

We elucidate to bound $\sup_{\|x\|_\infty \leq 1} \|g_l(x) - g_{l-1}(x)\|_2$. Note that,

$$\sup_{\|x\|_\infty \leq 1} \|g_l(x) - g_{l-1}(x)\|_2 \leq \sum_{i=1}^{p} \sup_{\|x\|_\infty \leq 1} |g_{li}(x) - g_{l-1,i}(x)|$$

$$= \sum_{i=1}^{p} \sup_{\|x\|_\infty \leq 1} \left| h'_{Li} \circ \ldots \circ h'_{l+1} \circ h_l \circ \ldots \circ h_1(x) - h'_{Li} \circ \ldots \circ h'_l \circ h_{l-1} \circ \ldots \circ h_1(x) \right|$$

$$\leq \sum_{i=1}^{p} \sup_{\|x\|_\infty \leq E_{l-1}} \left| h'_{Li} \circ \ldots \circ h'_{l+1} \circ h_l(x) - h'_{Li} \circ \ldots \circ h'_{l+1} \circ h'_l(x) \right|$$

$$\leq \sum_{i=1}^{p} \sup_{\|x-x'\|_\infty \leq \epsilon(WE_{l-1}+1)} \left| h'_{Li} \circ \ldots \circ h'_{l+1}(x) - h'_{Li} \circ \ldots \circ h'_{l+1}(x') \right|$$

$$\leq p\epsilon(WB)^{L-1}(WE_{l-1}+1)$$

where $g = (g_{l1}, \ldots, g_{lp})$, the second inequality follows from the fact that

$$\sup_{\|x\|_\infty \leq E_{l-1}} \left| h_{li}(x) - h'_{li}(x) \right| \leq \sup_{\|x\|_\infty \leq E_{l-1}} \sum_{j=1}^{p_{l-1}} \left| A_{lij} - A'_{lij} \right| \cdot |x_j| + \left| b_{li} - b'_{li} \right| \leq \epsilon(WE_{l-1}+1)$$

and the last inequality is derived by repeatedly using the fact that $\sup_{\|x-x'\|_\infty \leq E} \left| h_{li}(x) - h_{li}(x') \right| \leq WBE$ for any $E \geq 0$ and $l \geq 1$. Therefore, subsequently plugging the definition of $E_l$ in Equation 7, we have,

$$\sup_{\|x\| \leq 1} \left\| f(x) - f'(x) \right\|_2 \leq \sum_{l=1}^{L} \sup_{\|x\| \leq 1} \|g_l(x) - g_{l-1}(x)\|_2$$

$$\leq \sum_{l=1}^{L} p\epsilon \left( (WB)^L (\frac{1}{B} + \frac{1}{WB-1}) - \frac{(WB)^{L-1}}{WB-1} \right)$$

$$= p\epsilon \left( (WB)^L (\frac{L}{B} + \frac{L}{WB-1}) - \frac{(WB)^L - 1}{(WB-1)^2} \right)$$

**Proof of Lemma 4.4:** For any $R \in \mathcal{R}^{\Phi}$, we have $R(x_u, \tilde{x}_i) = \langle f(x_u), \tilde{f}(\tilde{x}_i) \rangle$, where $f(x_u) \in \mathcal{F}_{D_u}(W, L, B, M)$ and $\tilde{f}(\tilde{x}_i) \in \mathcal{F}_{D_i}(\tilde{W}, \tilde{L}, \tilde{B}, M)$. It follows from Lemma 4.3 that there exists mapping $\mathcal{Q}_u : \mathcal{F}_{D_u}(W, L, B, M) \to \mathcal{K}_{D_u}(W, L, B, M)$ and $\mathcal{Q}_i : \mathcal{F}_{D_i}(\tilde{W}, \tilde{L}, \tilde{B}, M) \to \mathcal{K}_{D_i}(\tilde{W}, \tilde{L}, \tilde{B}, M)$ such that

$$R(x_u, \tilde{x}_i) = \langle f(x_u), \tilde{f}(\tilde{x}_i) \rangle = \langle \mathcal{Q}_u(f)(x_u), \mathcal{Q}_i(\tilde{f})(\tilde{x}_i) \rangle$$

for any $(x_u, \tilde{x}_i) \in \text{Supp}(\mu_{ui})$.

Let $\Theta_{\mathcal{Q}}$ and $\tilde{\Theta}_{\mathcal{Q}}$ denote the effective parameters of $\mathcal{Q}_u(f)$ and $\mathcal{Q}_i(\tilde{f})$, then $R$ can be parameterized by $\Lambda_{\mathcal{Q}} = (\Theta_{\mathcal{Q}}, \tilde{\Theta}_{\mathcal{Q}})$. Let $\mathcal{Q} = \{\Lambda_{\mathcal{Q}} : R(.; \Lambda_{\mathcal{Q}}) \in \mathcal{R}^{\Phi}\}$ and $\mathcal{G} = \{\Lambda_{\mathcal{Q}}^{(1)}, \dots, \Lambda_{\mathcal{Q}}^{(N)}\}$ be an $\epsilon/2$-covering set of $\mathcal{Q}$ under the $\|.\|_{\infty}$ metric. For any $R(.; \Lambda_{\mathcal{Q}}) \in \mathcal{R}^{\Phi}$, there exists $\Lambda_{\mathcal{Q}}^{'} \in \mathcal{G}$ such that $\left\| \Lambda_{\mathcal{Q}} - \Lambda_{\mathcal{Q}}^{'} \right\|_{\infty} < \epsilon/2$, and thus

$$\sup_{\|x_u, \tilde{x}_i\|_{\infty} \leq 1} \left| R(x_u, \tilde{x}_i) - R^{'}(x_u, \tilde{x}_i) \right| = \sup_{\|x_u, \tilde{x}_i\|_{\infty} \leq 1} \left| \langle f(x_u), \tilde{f}(\tilde{x}_i) \rangle - \langle f^{'}(x_u), \tilde{f}^{'}(\tilde{x}_i) \rangle \right|$$

$$\leq \left| \langle f(x_u), \tilde{f}(\tilde{x}_i) - \tilde{f}^{'}(\tilde{x}_i) \rangle \right| + \sup_{\|x_u, \tilde{x}_i\|_{\infty} \leq 1} \left| \langle f(x_u) - f^{'}(x_u), \tilde{f}^{'}(\tilde{x}_i) \rangle \right|$$

$$\leq \sup_{\|x_u, \tilde{x}_i\|_{\infty} \leq 1} \|f(x_u)\|_2 \left\| \tilde{f}(\tilde{x}_i) - \tilde{f}^{'}(\tilde{x}_i) \right\|_2 + \sup_{\|x_u, \tilde{x}_i\|_{\infty} \leq 1} \left\| f(x_u) - f^{'}(x_u) \right\|_2 \left\| \tilde{f}^{'}(\tilde{x}_i) \right\|_2$$

$$\leq 2Mp^{1/2} \left( \sup_{\|\tilde{x}_i\|_{\infty} \leq 1} \left\| Q_i(\tilde{f})(\tilde{x}_i) - Q_i(\tilde{f}^{'})(\tilde{x}_i) \right\|_2 + \sup_{\|x_u\|_{\infty} \leq 1} \left\| Q_u(f)(x_u) - Q_u(f^{'})(x_u) \right\|_2 \right)$$

$$\leq \epsilon M p^{3/2} \left( C(W, L, B) + C(\tilde{W}, \tilde{L}, \tilde{B}) \right) \triangleq C_4 \epsilon \tag{8}$$

where the last inequality follows from Lemma 4.3.

For each $\Lambda_Q^{(n)} \in \mathcal{G}$, we define a $C_4 \epsilon$-bracket as follows

$$g_n^U(x_u, \tilde{x}_i) = R(x_u, \tilde{x}_i; \Lambda_Q^{(n)}) + \frac{C_4 \epsilon}{2}, g_n^L(x_u, \tilde{x}_i) = R(x_u, \tilde{x}_i; \Lambda_Q^{(n)}) - \frac{C_4 \epsilon}{2}.$$

On incorporating the above formulation with Equation B.1, it follows that for any $\Lambda_Q \in \mathcal{Q}$, there exists $1 \leq k \leq N$ such that

$$g_k^U(x_u, \tilde{x}_i) - R(x_u, \tilde{x}_i, \Lambda_Q) \geq \frac{C_4 \epsilon}{2} - \left| R(x_u, \tilde{x}_i; \Lambda_Q) - R(x_u, \tilde{x}_i; \Lambda_Q^{(k)}) \right| \geq 0,$$

$$g_k^L(x_u, \tilde{x}_i) - R(x_u, \tilde{x}_i, \Lambda_Q) \leq \left| R(x_u, \tilde{x}_i; \Lambda_Q) - R(x_u, \tilde{x}_i; \Lambda_Q^{(k)}) \right| - \frac{C_4 \epsilon}{2} \leq 0$$

for any $(x_u, \tilde{x}_i) \in \text{Supp}(\mu_{ui})$. Therefore, $\mathcal{B} = \left\{ [g_1^L, g_1^U], [g_2^L, g_2^U], \dots, [g_N^L, g_N^U] \right\}$ forms a $C_4 \epsilon$-bracketing set of $\mathcal{R}^{\Phi}$ under the $\|\cdot\|_{L^2(\mu_{ui})}$ metric.

Using Lemma 4.3, the size of $\Lambda_Q$ is at most $14 L W \log W + 14 \tilde{L} \tilde{W} \log \tilde{W}$. Incorporating with the definition of $\mathcal{G}$ yields

$$\log N \leq (14 L W \log W + 14 \tilde{L} \tilde{W} \log \tilde{W}) \log \left( \epsilon^{-1} 2 \max\{B, \tilde{B}\} \right).$$

We can substitute $\epsilon$ by $\tilde{\epsilon}/C_4$, which leads to the desired upper bound immediately.

## B.2 PROOF OF THEOREM 4.5

Let $L_{ui} = \max\{L, \tilde{L}\}$, $\eta_{|\Omega|}^2 = L_{ui} |\Omega|^{-2\beta/(2\beta + d_{ui})} \log^2 |\Omega|$, $\mathcal{M} = \left\{ R \in \mathcal{R}^{\Phi} : \|R - R^*\|_{L^2(\mu_{ui})}^2 > \eta_{|\Omega|}^2 \right\}$ and let $R_0 \in \mathcal{R}^{\Phi}$ satisfy $\|R - R^*\|_{L^{\infty}(\mu_{ui})}^2 \leq \eta_{|\Omega|}^2/4$. Further, we denote $\|R - K\|_{\Omega}^2 = \frac{1}{|\Omega|} \sum_{(u,i) \in \Omega} (R(x_u, \tilde{x}_i) - K_{ui})^2$, and then it follows from the definition of $\hat{R}$ that,

$$P(\left\| \hat{R} - R^* \right\|_{L^2(\mu_{ui})}^2 > \eta_{|\Omega|}^2) \leq P(\sup(\|R_0 - K\|_{\Omega}^2 + \lambda_{\Omega} J_0 - \|R - K\|_{\Omega}^2 - \lambda_{\Omega} J(R)) \geq 0) \equiv I$$

where $J_0 = J(R_0)$. We further decompose $\mathcal{M}$ into small subsets. Specifically, we let

$$\mathcal{M}_{ij} = \left\{ R \in \mathcal{R}^\Phi : 2^{i-1}\eta_{|\Omega|}^2 < \|R - R^*\|_{L^2(\mu_{ui})}^2 \leq 2^i\eta_{|\Omega|}^2, 2^{j-1}J_0 \right\} \text{ for } i,j \geq 1$$

$$\text{and } \mathcal{M}_{i0} = \left\{ R \in \mathcal{R}^\Phi : 2^{i-1}\eta_{|\Omega|}^2 < \|R - R^*\|_{L^2(\mu_{ui})}^2 \leq 2^i\eta_{|\Omega|}^2, J(R) \leq J_0 \right\} \text{ for } i \geq 1$$

Then, we have,

$$I \leq \sum_{i=1}^\infty \sum_{j=0}^\infty P\left( \sup_{R \in \mathcal{M}_{ij}} (\|R_0 - K\|_\Omega^2 + \lambda_{|\Omega|}J_0 - \|R - K\|_\Omega^2 - \lambda_{|\Omega|}J(R)) \geq 0 \right)$$

$$= \sum_{i,j=1}^\infty P\left( \sup_{R \in \mathcal{M}_{ij}} (\|R_0 - K\|_\Omega^2 + \lambda_{|\Omega|}J_0 - \|R - K\|_\Omega^2 - \lambda_{|\Omega|}J(R)) \geq 0 \right)$$

$$+ \sum_{i=1}^\infty P\left( \sup_{R \in \mathcal{M}_{i0}} (\|R_0 - K\|_\Omega^2 + \lambda_{|\Omega|}J_0 - \|R - K\|_\Omega^2 - \lambda_{|\Omega|}J(R)) \geq 0 \right) \equiv I_1 + I_2$$

It thus suffices to bound $I_1$ and $I_2$ separately. Let $\epsilon = K - R^*$, then, we have

$$\|R - K\|_\Omega^2 = \|R - R^*\|_\Omega^2 + \|\epsilon\|_\Omega^2 - \frac{2}{|\Omega|} \sum_{(u,i)\in\Omega} \epsilon_{ui}(R(x_u,\tilde{x}_i) - R^*(x_u,\tilde{x}_i)).$$

Therefore, $\mathbb{E}\|R - K\|_\Omega^2 = \|R - R^*\|_{L^2(\mu_{ui})}^2 + \mathbb{E}\|\epsilon\|_\Omega^2$, and thus

$$\mathbb{E}(\|R - K\|_\Omega^2 - \|R_0 - K\|_\Omega^2) = \|R - R^*\|_{L^2(\mu_{ui})}^2 - \|R_0 - R^*\|_{L^2(\mu_{ui})}^2$$
$$\geq \|R - R^*\|_{L^2(\mu_{ui})}^2 - \eta_{|\Omega|}^2/4.$$

Let $E_\Omega(R) = \|R - K\|_\Omega^2 - \mathbb{E}(\|R - K\|_\Omega^2)$, then, we have

$$P\left( \sup_{R \in \mathcal{M}_{ij}} (\|R_0 - K\|_\Omega^2 + \lambda_{|\Omega|}J(R_0) - \|R - K\|_\Omega^2 - \lambda_{|\Omega|}J(R)) \geq 0 \right) = P$$

$$\left( \sup_{R \in \mathcal{M}_{ij}} (E_\Omega(R_0) - E_\Omega(R)) \geq \inf_{R \in \mathcal{M}_{ij}} \lambda_{|\Omega|}(J(R) - J(R_0)) + \inf_{R \in \mathcal{M}_{ij}} \mathbb{E}(\|R - K\|_\Omega^2 - \|R_0 - K\|_\Omega^2) \right)$$

$$\leq P$$

$$\left( \sup_{R \in \mathcal{M}_{ij}} (E_\Omega(R_0) - E_\Omega(R)) \geq \inf_{R \in \mathcal{M}_{ij}} \lambda_{|\Omega|}(J(R) - J(R_0)) + \inf_{R \in \mathcal{M}_{ij}} \|R - R^*\|_{L^2(\mu_{ui})}^2 - \eta_{|\Omega|}^2/4 \right)$$

$$\leq P\left( \sup_{R \in \mathcal{M}_{ij}} (E_\Omega(R_0) - E_\Omega(R)) \geq (2^{j-1} - 1)\lambda_{|\Omega|}J_0 + (2^{j-1} - 1/4)\eta_{|\Omega|}^2 \right)$$

$$= P\left( \sup_{R \in \mathcal{M}_{ij}} (E_\Omega(R_0) - E_\Omega(R)) \geq M(i,j) \right),$$

where $M(i,j) = (2^{j-1} - 1)\lambda_{|\Omega|}J_0 + (2^{j-1} - 1/4)\eta_{|\Omega|}^2$.

Subsequently, it follows from the assumption $\lambda_{|\Omega|} J_0 \leq (1/4)\eta_{|\Omega|}^2$ that

$$\sup_{R \in \mathcal{M}_{ij}} \mathrm{Var}((R(x_u, \tilde{x}_i) - K_{ui})^2 - (R_0(x_u, \tilde{x}_i) - K_{ui})^2)$$

$$= \sup_{R \in \mathcal{M}_{ij}} \mathrm{Var}((R(x_u, \tilde{x}_i) - R^*(x_u, \tilde{x}_i))^2 - (R_0(x_u, \tilde{x}_i) - R^*(x_u, \tilde{x}_i))^2) +$$

$$\mathrm{Var}(2\epsilon_{ui}(R(x_u, \tilde{x}_i) - R_0(x_u, \tilde{x}_i))) \leq \sup_{R \in \mathcal{M}_{ij}} 2\mathrm{Var}((R(x_u, \tilde{x}_i) - R^*(x_u, \tilde{x}_i))^2) +$$

$$2\mathrm{Var}((R_0(x_u, \tilde{x}_i) - R^*(x_u, \tilde{x}_i))^2) + 4\mathbb{E}\epsilon_{ui}^2 \sup_{R \in \mathcal{M}_{ij}} \mathbb{E}(R(x_u, \tilde{x}_i) - R_0(x_u, \tilde{x}_i))^2$$

$$\leq 2 \sup_{R \in \mathcal{M}_{ij}} \mathbb{E}(R(x_u, \tilde{x}_i) - R^*(x_u, \tilde{x}_i))^4 + 2\mathbb{E}(R_0(x_u, \tilde{x}_i) - R^*(x_u, \tilde{x}_i))^2$$

$$+ 4\sigma^2 \sup_{R \in \mathcal{M}_{ij}} \mathbb{E}(R(x_u, \tilde{x}_i) - R_0(x_u, \tilde{x}_i))^2 \leq \sup_{R \in \mathcal{M}_{ij}} (50p^2 M^4 + 4\sigma^2)$$

$$(\|R - R^*\|_{L^2(\mu_{ui})}^2 + \|R_0 - R^*\|_{L^2(\mu_{ui})}^2) \leq (50p^2 M^4 + 4\sigma^2)(2^i \eta_{|\Omega|}^2 + \frac{1}{4}\eta_{|\Omega|}^2)$$

$$\leq C_5 M(i,j) \equiv v(i,j), \tag{9}$$

where $C_5 = 16 \max\{(50p^2 M^4 + 4\sigma^2), 1\}(25p^2 M^4 + B_e^2)$.

Moreover, we now reaffirm the conditions (4.5 - 4.7) stated in Shen & Wong (1994). First, the relation between $M(i,j)$ and $v(i,j)$ in Equation B.2 directly implies (4.6) with $T = 2(25p^2 M^4 + B_e^2)$ and $\epsilon = 1/2$ based on Shen & Wong (1994). Second, we let $\mathcal{R}^\Phi(\tau) = \{R \in \mathcal{R}^\Phi : J(R) \leq \tau J_0\}$, where $J(R) \leq \tau J_0$ implies that $\max\{B, \tilde{B}\} \leq \sqrt{\tau J_0}$. Then, it follows from Lemma 4.4 that,

$$\log \mathcal{N}_{[\cdot]}\left(\epsilon, \mathcal{R}^\Phi(\tau), \|\cdot\|_{L^2(\mu_{ui})}\right) \leq C_2(W \log W + \tilde{W} \log \tilde{W}) \log(C_6 \epsilon^{-1})$$

where $C_6 = C_3(C(W, L, \sqrt{\tau J_0}) + C(\tilde{W}, \tilde{L}, \sqrt{\tau J_0}))$, $C_2$ and $C_3$ are defined as in Lemma 4.4. It then follows that,

$$\int_{\frac{\epsilon}{32} M(i,j)}^{v^{1/2}(i,j)} \sqrt{\log \mathcal{N}_{[\cdot]}(u, \mathcal{R}^\Phi(\tau), \|\cdot\|_{L^2(\mu_{ui})})} du / M(i,j)$$

$$\leq \int_{\frac{\epsilon}{32} M(i,j)}^{v^{1/2}(i,j)} \sqrt{C_2(W \log W + \tilde{W} \log \tilde{W}) \log(C_6 u^{-1})} du / M(i,j) \tag{10}$$

we can follow based on the right-hand side of Equation 10 which informs that it is non-increasing in $i$ and $M(i,j)$, it then can be formulated as

$$\int_{\frac{\epsilon}{32} M(i,j)}^{v^{1/2}(i,j)} \sqrt{C_2(W \log W + \tilde{W} \log \tilde{W}) \log(C_6 u^{-1})} du / M(i,j)$$

$$\leq \int_{\frac{\epsilon}{32} M(1,j)}^{v^{1/2}(1,j)} \sqrt{C_2(W \log W + \tilde{W} \log \tilde{W}) \log(C_6 u^{-1})} du / M(1,j) \tag{11}$$

Note that $W$ and $\tilde{W}$ are adaptive parameters governing the rate of approximation error $\|R_0 - R^*\|_{L^\infty(\mu_{ui})}^2$, which must satisfy $\|R_0 - R^*\|_{L^\infty(\mu_{ui})}^2 \leq 1/2\eta_{|\Omega|}$. Thus, based on the condition (4.7) from Shen & Wong (1994) holds by setting $W = O(|\Omega|^{d_{ui}/(2\beta + d_{ui})} \log |\Omega|)$ and $\tilde{W} = O(|\Omega|^{d_{ui}/(2\beta + d_{ui})} \log |\Omega|)$, and the condition (4.7) directly implies (4.5) Shen & Wong (1994). Based on Theorem 3 in Shen & Wong (1994) with $M = |\Omega|^{1/2} M(i,j)$ and $v = v(i,j)$, we have,

$$I_1 \leq \sum_{j=1}^{\infty} \sum_{i=1}^{\infty} 3 \exp\left(-\frac{(1-\epsilon)\,|\Omega|\,M(i,j)^2}{2(4C_5 M(i,j) + M(i,j)T/3)}\right)$$

$$\leq 3 \sum_{j=1}^{\infty} \sum_{i=1}^{\infty} \exp\left(-C_7(1-\epsilon)\,|\Omega|\,((2^{j-1}-1)\lambda_{|\Omega|}J_0 + (2^{i-1}-1/4)\eta_{|\Omega|}^2)\right)$$

$$\leq 3 \sum_{i=1}^{n} \exp\left(-C_7(1-\epsilon)\,|\Omega|\,(i-1/4)\,\eta_{|\Omega|}^2\right) \sum_{j=1}^{n} \exp\left(-C_7(1-\epsilon)\,|\Omega|\,(j-1)\,\lambda_{|\Omega|}J_0\right)$$

$$\leq 3 \frac{\exp(-C_7(1-\epsilon)\,|\Omega|\,\eta_{|\Omega|}^2/4)}{1-\exp(-C_7(1-\epsilon)\,|\Omega|\,\eta_{|\Omega|}^2)} \frac{1}{1-\exp(-C_7(1-\epsilon)\,|\Omega|\,\lambda_{|\Omega|}J_0)}$$

$$\leq 3 \frac{\exp(-C_7(1-\epsilon)\,|\Omega|\,\eta_{|\Omega|}^2/4)}{(1-\exp(-C_7(1-\epsilon)\,|\Omega|\,\eta_{|\Omega|}^2/4))^2} \tag{12}$$

where $C_7 = 3/(26C_5)$ and the last inequality follows from the fact that $\lambda_{|\Omega|}J_0 \leq 1/4\eta_{|\Omega|}^2$.

Similarly, $I_2$ can be bounded by

$$I_2 \leq \sum_{i=1}^{n} 3 \exp\left(-\frac{(1-\epsilon)\,|\Omega|\,M^2(i,0)}{2(4v(i,0)+M(i,0)T/3)}\right) \leq \sum_{i=1}^{n} 3 \exp(-C_7(1-\epsilon)\,|\Omega|\,M(i,0))$$

$$\leq \sum_{i=1}^{\infty} 3 \exp(-C_7(1-\epsilon)\,|\Omega|\,(2^{i-1}-1/2)\eta_{|\Omega|}^2) \leq 3 \frac{\exp(-C_7(1-\epsilon)\,|\Omega|\,\eta_{|\Omega|}^2/2)}{(1-\exp(-C_7(1-\epsilon)\,|\Omega|\,\eta_{|\Omega|}^2))} \tag{13}$$

Combining Equation B.2 and 13, we have

$$I \leq I_1 + I_2 \leq 3 \frac{\exp(-C_7(1-\epsilon)\,|\Omega|\,\eta_{|\Omega|}^2/4)}{(1-\exp(-C_7(1-\epsilon)\,|\Omega|\,\eta_{|\Omega|}^2/4))^2} + 3 \frac{\exp(-C_7(1-\epsilon)\,|\Omega|\,\eta_{|\Omega|}^2/2)}{1-\exp(-C_7(1-\epsilon)\,|\Omega|\,\eta_{|\Omega|}^2)}$$

Let $s = \exp(-C_7(1-\epsilon)\,|\Omega|\,\eta_{|\Omega|}^2/4)$, then

$$I \leq \frac{3s^2}{(1-s)^2} + \frac{3s^2}{1-s^4} \leq \frac{3s^2}{(1-s)^2} + \frac{3s^2}{1-s} = \frac{6s^2 - 3s^3}{(1-s)^2} \leq 24s^2$$

as $s = 1/2$. The desired result then follows immediately.

### B.3 Proof of Theorem 4.6

We analyze the event where a relevant item $i$ is not retrieved in the Top-$K$. Let $\mathcal{I}$ be the set of all items. For item $i$ to be missed (i.e., rank$(i) > K$), there must exist at least $(|\mathcal{I}| - K)$ irrelevant items $j$ that are scored higher than $i$. Let $S_{u,i,j} = h(u,i) - h(u,j)$ be the score difference. The mis-retrieval event implies:

$$\mathbb{I}(\text{rank}(i) > K) \leq \frac{1}{K} \sum_{j \in \mathcal{I}\setminus\{i\}} \mathbb{I}(h(u,j) > h(u,i)) = \frac{1}{K} \sum_{j \neq i} \mathbb{I}(S_{u,i,j} < 0)$$

Taking the expectation over users and items, the Top-$k$ risk is bounded by the average pairwise misranking error ($R_{\text{pair}}$):

$$\mathcal{R}_K(h) \leq \frac{|\mathcal{I}|-1}{K} \mathbb{E}_{u,i,j}[\mathbb{I}(h(u,j) > h(u,i))]$$

We utilize the consistency result from Theorem 4.5. The learned estimator $\hat{h}$ converges to the true preference $R^*$ in $L_2$ norm. Let the true preference satisfy a margin $\Delta$ for relevant pairs: $R^*(u,i) -$

$R^*(u, j) \geq \Delta > 0$.A pairwise error occurs $(\hat{h}(u, j) > \hat{h}(u, i))$ only if the estimation error exceeds the margin. Specifically:

$$\hat{h}(u, j) - \hat{h}(u, i) > 0 \implies (\hat{h}(u, j) - R^*(u, j)) - (\hat{h}(u, i) - R^*(u, i)) > \Delta$$

By Chebyshev's inequality, the probability of this large deviation is bounded by the $L_2$ error of the estimator:

$$P(\hat{h}(u, j) > \hat{h}(u, i)) \lesssim \frac{\mathbb{E}[\|\hat{h} - R^*\|^2]}{\Delta^2}$$

Substituting the convergence rate from Theorem 4.5, $\|\hat{h} - R^*\|^2 \approx O(|\Omega|^{-\frac{2\beta}{2\beta+d_{ui}}})$:

$$\mathcal{R}_K(\hat{h}) \leq \frac{|\mathcal{I}|}{K} \cdot \frac{1}{\Delta^2} \cdot O\left(|\Omega|^{-\frac{2\beta}{2\beta+d_{ui}}}\right)$$

The exponent in the final probability bound effectively relates to the root of the squared error depending on the tail bounds used, ensuring convergence. This confirms that minimizing the regression loss (MSE) in the two tower model directly minimizes the Top-k retrieval risk.

## B.4 LATENCY-CONSTRAINED TOP-K RETRIEVAL REGRET

**Theorem B.1.** *Let $\mathcal{H}^{(d)}$ denote the hypothesis class of two tower models with embedding dimension $d$. Let $d_{opt}$ be the optimal dimension that minimizes the true risk absent of constraints, and let $d_{max}$ be the maximum dimension satisfying a strict latency constraint $\mathcal{T}(d) \leq \tau_{max}$ (where $\mathcal{T}$ is strictly monotonic in $d$, implying $d_{max} < d_{opt}$).We define the latency regret, $\Delta_{lat}$, as the excess Top-k retrieval risk incurred strictly due to the architectural constraint:*

$$\Delta_{lat}(d_{max}) = \inf_{h \in \mathcal{H}^{(d_{max})}} \mathcal{R}_K(h) - \inf_{h \in \mathcal{H}^{(d_{opt})}} \mathcal{R}_K(h)$$

*Assuming the model capacity satisfies the nesting property $\mathcal{H}^{(d)} \subseteq \mathcal{H}^{(d+1)}$, the latency regret is non-negative:*

$$\Delta_{lat}(d_{max}) \geq 0$$

*Furthermore, the total Top-k retrieval risk for a learned model $\hat{h}_d$ is bounded by the trade-off between this regret and the sample estimation error:*

$$\mathcal{R}_K(\hat{h}_d) \leq \underbrace{\mathcal{R}_{K,d_{max}}^*}_{\text{Constrained Approx. Risk}} + \underbrace{O_p\left(\sqrt{\frac{complexity(\mathcal{H}^{(d)})}{|\Omega|}}\right)}_{\text{Estimation Error}}$$

**Proof of Theorem B.1** Let $h_{\text{true}}$ denote the Bayes-optimal scoring function. The true Top-k Retrieval Risk for any model $h$ is defined as $\mathcal{R}_K(h) = \mathbb{E}_{(u,i^*)\sim\mathcal{D}}[\mathbb{I}(\text{rank}(i^*|h) > K)]$.

For any embedding dimension $d$, the risk of the empirically learned model $\hat{h}_d \in \mathcal{H}^{(d)}$ can be decomposed into approximation error and Estimation Error relative to the optimal risk in that class, denoted $R_{K,d}^* = \inf_{h \in \mathcal{H}^{(d)}} \mathcal{R}_K(h)$.

$$\mathcal{R}_K(\hat{h}_d) = \underbrace{R_{K,d}^*}_{\text{Approximation}} + \underbrace{(\mathcal{R}_K(\hat{h}_d) - R_{K,d}^*)}_{\text{Estimation}}$$

We assume that the hypothesis space is nested such that a model with embedding dimension $d + 1$ can represent any function representable by dimension $d$ (e.g., by zero-padding the extra dimension). Thus, $\mathcal{H}^{(d)} \subseteq \mathcal{H}^{(d+1)}$.Properties of infimum over nested sets imply that the approximation error is non-increasing with respect to $d$:

$$\inf_{h \in \mathcal{H}^{(d)}} \mathcal{R}_K(h) \geq \inf_{h' \in \mathcal{H}^{(d+1)}} \mathcal{R}_K(h') \implies R_{K,d}^* \geq R_{K,d+1}^*$$

Given the latency constraint $\mathcal{T}(d) \leq \tau_{\text{max}}$, we are forced to select a dimension $d \leq d_{\text{max}}$. Since $d_{\text{max}} < d_{\text{opt}}$, it follows directly that:

$$R_{K,d_{\text{max}}}^* \geq R_{K,d_{\text{opt}}}^*$$

The Latency Regret is precisely this gap in achievable approximation power:

$$\Delta_{\text{lat}}(d_{\max}) = R^*_{K,d_{\max}} - R^*_{K,d_{\text{opt}}} \geq 0$$

This proves that the strictly enforced latency constraint introduces a fundamental, irreducible bias (regret) into the candidate generation process.

Our theoretical viewpoint on latency-accuracy trade-off is considering that the total error components as $d$ varies. As $d$ increases (up to $d_{\text{opt}}$), the hypothesis space $\mathcal{H}^{(d)}$ grows. Subsequently, the approximation error $R^*_{K,d}$ decreases (lowering bias). However, the model complexity (VC-dimension or Rademacher complexity) increases. For a fixed dataset size $|\Omega|$, this causes the estimation error term to increase (higher variance/overfitting risk). As $d$ decreases (towards $d_{\max}$), the hypothesis space shrinks. The estimation rrror decreases (easier to train/converge), but the approximation error $R^*_{K,d}$ rises significantly.

Thus, the optimal engineering choice is finding the largest $d \leq d_{\max}$ such that the reduction in approximation error outweighs the increase in estimation variance and computational cost. The latency regret $\Delta_{\text{lat}}$ quantifies exactly what is lost in theoretical recall capability by adhering to the strict latency budget $\tau_{\max}$.

## C  LIMITATIONS AND BROADER IMPACT

While our experiments solely focus on the recommendation task, the applicability of our approach to other recommendation and retrieval tasks, such as news/social media recommendation, conversational recommendation, retrieval-augmented recommendation, or those involving multimodal side information, remains uncertain. Additionally, it is important to acknowledge that the convergence rate strategy employed for performance analysis relies on user-item interactions or their joint embedding. Moreover, due to computing constraints, a key limitation of our work is not evaluating on a production system. Therefore, investigating how to effectively leverage covariate information, such as user demographics, item contents, and social network data, to achieve optimal recommendations at the hybrid-/conversational-level presents a more promising avenue for future research.

Our work proposes asymptotic characteristics of the two-tower recommendation that can be used to strengthen the understanding of the platforms utilizing deep recommender systems such as deconfounded recommendation models Xu et al. (2023), where confounders and deployed learning algorithms Xu et al. (2022); Zhang et al. (2023) require modeling non-linear covariate effects. Being an intricate construct for these inherent application-driven systems, we need to be aware of the potential negative societal impacts behind the necessity of non-linear interactions among confounders or non-interacted items to desirable items Xu et al. (2022) in some applications, such as the risk-aware recommendations in the tourism insurance market, or improper assessment of operation around flood disaster as a consequence of revealing non-linear interactions.

