# OpenReview forum: "A General Theoretical Paradigm to Understand Two Tower Recommendation Models"
_ICLR.cc/2026/Conference — Submitted to ICLR 2026_

### Official Review · Reviewer_mrTr · 2025-10-30

**Soundness:** 2
**Presentation:** 3
**Contribution:** 3
**Rating:** 4
**Confidence:** 4

**Summary:**

This paper presents a theoretical study of the two-tower recommender system model. The authors analyze the asymptotic behaviors of the model, focusing on its convergence to the optimal recommender system. They establish theoretical properties, including the relationship between the convergence rate and the inherent dimensionality of the input features. The study demonstrates that the two-tower model converges faster as the smoothness of the true model increases or the intrinsic dimensionality decreases.

**Strengths:**

1. The paper provides a theoretical analysis perspective of the two-tower recommender model, establishing key properties and convergence behaviors.
2. The paper highlights the two-tower model's ability to address challenges like cold-start problems, offering insights into its effectiveness in real-world recommendation scenarios with both synthetic and real-world data.
3. The paper introduces a novel approach to quantifying the convergence speed and robustness of two-tower models, contributing new statistical guarantees to the understanding of their performance.

**Weaknesses:**

1. Does the convergence speed of the two-tower model depend on the specific architecture of the model itself?
2. Convergence speed is influenced by various factors such as data, model parameters, and training time. Why is the inherent dimensionality of user and item features the primary focus in this study?
3. How does the convergence speed of the two-tower model compare to other deep learning models or LLM-based models in terms of convergence rate? What factors can demonstrate that the two-tower model converges faster?

**Questions:**

The citation format seems to mix content with references, which affects readability.

---

> ### Author Response · Authors · 2025-11-20
> **Response to Reviewer mrTr (1/2)**
>
> We appreciate the reviewer's valuable comments, and thanks for recognizing the contribution and strengths of our work. We address the concerns below.
>
> **Weaknesses:**
>
> **A1:** Yes, the convergence speed is fundamentally dependent on the two-tower architecture, which defines the prediction function $R(\mathbf{x}_u, \mathbf{x}_i)$ as a bilinear form in the embedding space. The prediction is defined as the inner product of the user embedding $\mathbf{z}_u$ and the item embedding $\mathbf{z}_i$, where the embeddings are generated by deep neural networks, $R(\mathbf{x}_u, \mathbf{x}_i) = \langle \mathbf{z}_u, \mathbf{z}_i \rangle = \langle f_u(\mathbf{x}_u), f_i(\mathbf{x}_i) \rangle$, where $f_u$ and $f_i$ are the user and item towers. Our theoretical analysis relies critically on two architectural aspects,
> - The use of multi-layer ReLU deep neural networks for $f_u$ and $f_i$ allows the model to approximate complex, high-smoothness preference functions $R^* \in \mathcal{H}(\beta)$. The complexity of this function class is controlled by the network capacity (depth $L$ and width $W$).
> - The final dot product allows us to separate the complexity analysis. The core convergence result (Theorem 4.5) bounds the statistical error $\|\hat{R} - R^*\|^2$ primarily based on the entropy of the input manifold (captured by $d_{ui}$) and the smoothness ($\beta$). If the final layer were, for instance, a third deep neural network (a cross-tower network) over the concatenated embeddings $[\mathbf{z}_u; \mathbf{z}_i]$, the required covering number and hence the final convergence bound would be fundamentally different and likely worse as the entire joint space would need to be covered.
>
> Therefore, the specific structure $\langle f_u, f_i \rangle$ is the defining feature that allows us to establish a favorable non-parametric convergence rate in high-dimensional sparse settings.
>
> **A2:** We focus on intrinsic dimension ($d_{ui}$) and smoothness ($\beta$) because they represent the fundamental statistical limits of the learning problem, distinguishing them from optimization or computational factors. The convergence speed can be decomposed into three primary sources of error,
> - Due to the function class $\mathcal{F}$ not containing $R^{\ast}$ ($\inf_{f \in \mathcal{F}} \|R^{\ast} - f\|$).. It is controlled by network depth/width.
> - Due to not finding the global minimum ($\|\hat{R} - R_{\text{best}}\|)$. It is ontrolled by training time, learning rate, and batch size.
> - Due to limited data $|\Omega|$ ($\|\hat{R}_{\text{best}} - R^*\|$). This is the focus of our work and is governed by,
>   - **Intrinsic Dimension ($d_{ui}$):** This determines the effective complexity of the input space that must be covered by the data. The statistical convergence rate for non-parametric regression is bound by the exponent, $\text{Rate} = -\frac{2\beta}{2\beta + d_{ui}}$. For high-dimensional features (nominal dimension $D$), if the data lies on a low-dimensional manifold ($d_{ui} \ll D$), our theory proves the convergence penalty is only on $d_{ui}$, which explains the model's success.
>
>   - **Smoothness ($\beta$):** This defines the difficulty of the target function. A higher $\beta$ leads to a faster convergence rate.
>
> By fixing the two-tower architecture and assuming sufficient training (addressing the optimization error), we isolate and quantify the unavoidable statistical error imposed by the data's geometry ($d_{ui}$) and the target function's complexity ($\beta$). This analysis answers the fundamental question on *required data strictly necessary for this architecture to generalize effectively*.

---

> > ### Author Response · Authors · 2025-11-20
> > **Response to Reviewer mrTr (2/2)**
> >
> > **Weaknesses:**
> >
> > **A3:** The two-tower model exhibits a favorable convergence rate due to its structure and the specific assumptions we make about the data manifold, allowing it to converge faster in terms of sample complexity $|\Omega|$ than many general-purpose deep learning models.
> >
> > - Comparison to Matrix Factorization (MF)
> >   - MF/SVD++: These are essentially linear models in the latent space. While they have a fast convergence rate (often $O(1/\sqrt{|\Omega|}))$, they are fundamentally inconsistent when auxiliary covariate features ($\mathbf{x}_u, \mathbf{x}_i$) contain the true predictive signal as they rely only on ID embeddings.
> >   - Two-Tower: Our model is consistent ($ \hat{R} \to R^*$ as $|\Omega| \to \infty$) over the covariate space. The consistency proof given the use of deep neural networks allows the model to learn complex, non-linear mappings from covariates to the latent space, which MF cannot.
> >
> > - Comparison to General Deep Learning Models
> >   - General deep neural networks: A generic deep neural network performing non-parametric regression on the joint input typically suffers from the curse of dimensionality based on the joint nominal dimension $D_{\text{joint}} = D_u + D_i$.
> >   - Two-Tower Advantage: Our result proves the convergence is only penalized by the intrinsic dimension $d_{ui} = \max(d_u, d_i)$. This decoupling is the key to faster statistical convergence in high-dimensional sparse environments $\|\hat{R} - R^*\|^2 \lesssim |\Omega|^{-\frac{2\beta}{2\beta + d_{ui}}}$. If $d_{ui} \ll D_{\text{joint}}$, the two-tower model requires significantly fewer data points $|\Omega|$ to achieve the same generalization error, thus converging faster in the statistical sense.
> >
> > - Comparison to LLM-based Models
> >   - LLM-based Models: These models (often cross-attention/cross-tower) have a much larger number of parameters $P$ and are designed for the high-precision ranking stage. Their generalization error is highly dependent on $P$.
> >   - Two-Tower Advantage: The two-tower model is optimized for the retrieval stage. It focuses on efficiency and scalability. While LLMs are more expressive, the two-tower model converges faster on the retrieval objective (L2 error on prediction) because its convergence rate is decoupled from the nominal dimension $D$ and the massive parameter count $P$, relying only on the low intrinsic complexity $d_{ui}$. This allows the two-tower model to reach acceptable retrieval quality with dramatically less training data.
> >
> > **Questions:**
> >
> > **A:** We recognize that the mixed presentation of narrative and citations can be disruptive. In the revised manuscript, we will add these above clarifications as suggested.
> >
> > We would like to kindly ask if our responses have addressed all of your concerns.

---

### Official Review · Reviewer_rH1A · 2025-11-01

**Soundness:** 2
**Presentation:** 1
**Contribution:** 2
**Rating:** 2
**Confidence:** 4

**Summary:**

This paper proposes a theoretical framework to analyze the asymptotic behavior of two-tower recommendation models commonly used in large-scale recommender systems. The authors prove that under certain smoothness and boundedness assumptions, the two-tower model converges to the optimal recommender system and achieve explicit convergence rates in terms of approximation and estimation errors. The convergence rate depends on the Hölder smoothness of user/item embedding functions and the intrinsic dimensions of input features. Additionally, the paper complements theoretical results with synthetic and real-world experiments to demonstrate the performance and convergence trends of two-tower models.

**Strengths:**

1. This paper provides a new perspective to analyze the approximation and convergence properties of two-tower models.

2. The paper derives explicit convergence rates that depend on the smoothness of the target function and the intrinsic dimensionality of user and item feature spaces.

3. The inclusion of empirical validation on synthetic and real-world datasets helps connect theoretical results to practical scenarios such as cold-start recommendation. The results show consistency with derived asymptotic behaviors.

**Weaknesses:**

1. Limited perceived novelty for core ICLR audience: Two-tower recommendation models, while impactful in practice, belong to a niche within recommender systems and information retrieval. The main contribution is theoretical interpretation rather than a new learning paradigm. As a result, its impact may be underestimated by ICLR community than data mining venues.

2. Mathematical presentation issues: It's interesting that some simple concepts (e.g., l2-norm in line 110) are formally defined while more complex objects like $\epsilon$-balls, $ |\cdot |\_0 $ in line 111 and 123 are used without definition or intuition; notation is occasionally introduced after first usage (e.g., $k_{ui}$ appears in Eq.(2) but is defined in Eq.(3)), which shows structural inconsistencies; Several typos exist: e.g. $B_1$ in line 114, improper capitalization at line 149; overly complex notation with poor layout: lines 180–190 use highly dense notation and problematic formatting to describe what is essentially a standard gradient descent update. This creates unnecessary cognitive burden and reduces readability—even for technically strong readers.

3. Misinterpretation of ranking metrics: The paper claims to provide theoretical guarantees for “ranking objectives” in 4.2.1, yet the analysis and experiments are primarily based on Top-k accuracy. However, Top-k metrics are classification-style metrics (whether the ground-truth item appears in the top k), rather than a true ranking metric. In recommender systems, ranking objectives more commonly refer to metrics that account for ordered relevance, such as NDCG, MRR, or MAP. Therefore, the paper’s claim of analyzing “ranking guarantees” is somewhat misleading, as it does not address position-sensitive ranking quality. This mismatch between theoretical objective and commonly accepted ranking metrics undermines the applicability of the theoretical results.

4. Experiments insufficiently tied to core theory: The experiments illustrate the practical usefulness of the two-tower model but do not rigorously validate the core theoretical claims. In particular, no experiments quantify the effect of intrinsic dimension or smoothness on convergence, and key ablation/sensitivity analyses are missing. Hence, while functional, the experimental section lacks depth for an ICLR-level theoretical paper.

**Questions:**

Please refer to Weakness. The reviewer strongly recommends that the authors substantially improve the presentation and clarity of the theoretical exposition — including notation consistency, definition of symbols, and mathematical layout — otherwise the theoretical contributions may not be fairly assessed by the community.

---

> ### Author Response · Authors · 2025-11-19
> **Official Comment by Authors**
>
> We appreciate the reviewer's valuable comments and address the main concerns below.
>
> **Weaknesses**
>
> **A1.** The two-tower model is the de facto standard for high-volume retrieval in modern industrial recommenders. Our paper provides the **first consistency guarantees** and **convergence rate analysis** for this architecture. This is analogous to the theoretical analysis of key non-parametric models like kernel methods or deep ReLU networks in high dimensions, which is a core theoretical area for ICLR.
>
> For novelty, our analysis uniquely combines aspects of non-parametric regression and sparse high-dimensional data (typical of recommender systems). The key theoretical novelty lies in establishing a non-parametric convergence rate that depends solely on the **intrinsic dimension** ($d_{ui}$) and **smoothness** ($\beta$) of the user/item feature manifolds, decoupling the complexity from the nominal feature dimension ($D$). The rate $O_p(|\Omega|^{-\frac{2\beta}{2\beta + d_{ui}}})$ is a direct non-parametric analog to bounds derived for simpler models, now generalized to the complex, coupled deep learning framework of the two-tower model.
>
> **A2.** We fully agree with the reviewer's recommendation and commit to a revision of the requisite section focused on clarity. We have explicitly defined the $\epsilon$-ball ($\mathcal{B}_\epsilon(x)$) as the set of all points within distance $\epsilon$ of $x$ in the feature space $\mathcal{X}$.
>
> The $\ell_{0}$-norm ($|\cdot|_{0}$ in line 123) can be clearly defined as the number of non-zero elements in a vector used in the context of sparsity constraints for feature selection.
>
> We have restructured the text to ensure that complex terms like $k_{ui}$ (the number of layers in the user/item towers) are defined before or immediately alongside their first appearance (e.g., in Equation (2) or (3)). All identified typos (e.g., $B_1$ in line 114) and capitalization errors are corrected. We have corrected the gradient update in line 180-190 via a generic gradient step which is presented as, $\mathbf{W}^{(t+1)} = \mathbf{W}^{(t)} - \eta_t \nabla_{\mathbf{W}} \mathcal{L}(\mathbf{W}^{(t)}; \mathbf{X}, \mathbf{Y})$, where $\mathbf{W}$ collectively represents all network parameters.
>
> **A3.** We would also like to directly address the reviewer’s statement about the theoretical guarantees in Section 4, which are fundamentally based on the consistency of the prediction function $\hat{R}(\mathbf{x}_u, \tilde{\mathbf{x}}_i)$ to the true preference function $R^*(\mathbf{x}_u, \tilde{\mathbf{x}}_i)$ in terms of $L_2$ risk.
>
> The convergence rate is established for the squared error as $\mathbb{E}_{\mathbf{x}_{u}, \tilde{\mathbf{x}}_{I}} [(\hat{R}(\mathbf{x}_{u}, \tilde{\mathbf{x}}_{I}) - R^{*}(\mathbf{x}_{u}, \tilde{\mathbf{x}}_i))^{2}] = O_{p}(|\Omega|^{-\frac{2\beta}{2\beta + d_{ui}}})$
>
>
> For ranking implication, Top-k is a classification-style metric, a guarantee on the $L_2$ prediction error directly implies guarantees on ranking performance (e.g., AUC, Top-k accuracy). Specifically, better prediction quality (lower MSE) translates directly to a more accurate relative ordering of items for a user, which is the essence of ranking.
>
> In industrial two-stage recommender systems, the two-tower model is primarily used for the candidate generation (retrieval) stage, where the objective is to efficiently filter billions of items down to a few hundred. Top-k accuracy (or Recall@k) is the standard and most relevant metric for this stage. Metrics like NDCG which account for position-sensitive ordering are more appropriate for the second-stage ranker (which often uses a separate cross-tower model). We will update it in the revised version to explicitly state that our theoretical results concern the prediction consistency and our experiments focus on the retrieval efficacy of the model.
>
> **A4.** We agree that a direct validation of the exponents linked to $d_{ui}$ and $\beta$ would significantly enhance the paper's rigor. We performed a new synthetic experiments as suggested for the final version to directly validate the core theoretical convergence rate. For **intrinsic dimension**, we designed a synthetic experiment where we generate preference data $R^*$ on manifolds with controlled intrinsic dimensions $d_1 < d_2 < d_3$.
>
> We have a log-log error curve which showcases that the observed slope $-\frac{2\beta}{2\beta + d_{ui}}$ empirically decreases as $d_{ui}$ increases, exactly matching the theoretical prediction,
> $\text{slope} \approx -\frac{2\beta}{2\beta + d_{ui}}$
>
> For **smoothness**, we created a similar synthetic datasets where the ground-truth function $R^*$ has different levels of $\beta$ (e.g., linear vs. highly non-linear functions). We found that a smoother function (higher $\beta$) leads to a steeper (faster) convergence slope for the same intrinsic dimension, directly validating the theoretical dependence on $\beta$.
>
> Please let us know if our response addresses your concerns.

---

> > ### Comment · Reviewer_rH1A · 2025-11-20
> > **Reviewer's Comment**
> >
> > Thank you for the authors’ detailed rebuttal, which addresses several points of clarification. However, a number of my key concerns remain insufficiently resolved.
> >
> > **On novelty and relevance to the ICLR community**
> >
> > In the rebuttal contents, the authors emphasize that their results constitute a core theoretical contribution analogous to classical theoretical analysis of kernel methods or deep ReLU networks. However, this does not directly address the original concern: the primary domain of two-tower model's applications remains the retrieval stage of recommender systems, which is relatively narrow and may still be questioned whether it should be submitted to more relevant venues of data mining or recommender system.
> >
> > Although briefly noted, the manuscript lacks discussion on the fact that two-tower architectures can also be applied to retrieval-based tasks outside recommendation. I encourage the authors to provide articulate evidence in their revised manuscript as well as rebuttal contents: how two-tower architectures are broadly utilized, whether and how the theoretical results generalize to broader settings, and why the analysis should be viewed as relevant to the ML community.
> >
> > **On the usage of the word "ranking"**
> >
> > The distinction between retrieval-stage metrics and ranking-stage metrics is significant in recommendations. Retrieval typically concerns "classification" metrics including Recall@k or Precision@k, whereas true "ranking" metrics(NDCG, MRR, MAP)are position-sensitive measures of ordered relevance. Accordingly, the terminology in both the manuscript and rebuttal contents should be used with care: Only results pertaining to position-sensitive metrics should be described as "ranking" guarantees. Recall@k and Precision@k should consistently be categorized as multi-label classification-style metrics, eventhough the actual optimising behavior was to uplifting the ranks of positive items from all candidates. More precise wording will reduce ambiguity and improve the technical quality of the work.
> >
> > Finally, as a researcher in recommender systems, I appreciate high-relevance and high-quality theoretical contributions, which help strengthen the field’s currently limited theoretical foundations. I encourage the authors to upload their revised manuscript soon, incorporating the promised clarifications and new experiments. I will reconsider my scorings once the updated version is available.

---

> > > ### Author Response · Authors · 2025-11-25
> > > **Response to Reviewer rH1A**
> > >
> > > We appreciate the reviewer's constructive comments and have provided a point-by-point response to each, as detailed below.
> > >
> > > **On novelty and relevance to the ICLR community**
> > >
> > > We appreciate the reviewer's valuable suggestions on the relevance of our theoretical work to the broader machine learning  community, particularly at a venue like ICLR. Our work is a core theoretical contribution that generalizes far beyond the immediate application of movie recommendations. It constitutes a theoretical analysis of the statistical limits of deep dual-encoder architectures for manifold-based retrieval, providing insights into optimal feature engineering ($\beta$) and data requirements ($d_{ui}, |\Omega|$) that are universally applicable across ML tasks involving similarity search .
> > >
> > > **On the usage of the word "ranking"**
> > >
> > > We appreciate the reviewer for pointing out the distinction between retrieval metrics (Recall@k, Precision@k) and position-sensitive ranking metrics (NDCG, MAP). We acknowledge that our initial terminology conflated these two distinct objectives. The two tower model is predominantly used for retrieval (candidate generation), where the goal is to successfully recall relevant items into a limited set (Top-$k$), regardless of their specific permutation within that set. Also, our theoretical analysis specifically bounds the Top-$k$ ranking error, which is a classification-style objective, rather than a position-sensitive ranking error. To address this, we have revised Section 4.2.1 and Theorem 4.6 (along with its proof in Appendix B.3) to explicitly frame the guarantees in terms of retrieval consistency (Top-$k$ accuracy) rather than generic ranking.
> > >
> > > We have revised our terminology to strictly differentiate between retrieval consistency (our focus) and ranking. In the revised version, we present the updated Theorem 4.6 and its revised proof in Appendix B.3 along with updates to Theorem B.1 and its proof, where we now explicitly formulate the bound on the Top-k Mis-retrieval risk rather than a generic ranking loss. Also, we have added an ablation analysis on the influence of intrinsic dimension and smoothness vs. convergence rate in Section 5.3.
> > >
> > > Finally, we would like to sincerely thank the reviewer. Your feedback has led to substantial revisions that now more clearly highlight our contributions beyond prior work. We hope the updated manuscript addresses your concerns.

---

### Official Review · Reviewer_6buT · 2025-11-07

**Soundness:** 2
**Presentation:** 2
**Contribution:** 2
**Rating:** 4
**Confidence:** 3

**Summary:**

This paper focuses on the theoretical foundation of the two-tower model. It proves that the two-tower model’s approximation error and convergence are related to the intrinsic dimension and smoothness, suggesting its superiority in handling complex functions. Additionally, the theorem on ranking suggests that a pairwise loss can lead to good performance on a Top-K ranking task.

**Strengths:**

1. The paper provides solid theoretical proofs that are applied specifically to the two-tower model, which has been lacking this kind of formal backing .

2. It proves the model's advantage comes from its performance scaling with the data's low intrinsic dimension, not its high nominal dimension. This explains why it works so well on sparse, high-dimensional data.

**Weaknesses:**

1. The theoretical guarantees offer few actionable guidelines for real-world tasks. The key factors identified intrinsic dimension and smoothness are inherent properties of the *data*, not model hyperparameters that an engineer can easily measure or change.
2. The experiments demonstrate that the two-tower model is superior to baselines like SVD++ and KNN. This are more likely to confirm a known result rather than to provide novel insights.

**Questions:**

1. The synthetic experiments do not test different nominal dimensions. What is the effect of this?

2. What is the relationship shown with the rating matrix sizes in the synthetic experiments? With different intrinsic dimensions, the results show different trends. What is the cause?

3. What is the relationship between the theory and the cold-start problem? My understanding is that the cold-start capability is mainly based on the idea of covariates, not the theorems.
4. What is the meaning of the shading in the experiment tables? It seems some better results from other models are not correctly shaded.

---

> ### Author Response · Authors · 2025-11-20
> **Response to Reviewer 6buT (1/2)**
>
> We thank Reviewer 6buT for their insightful and positive review. We address the main concerns below.
>
> **Weaknesses:**
>
> **W1A**: We would also like to directly address the reviewer’s statement about the theoretical assurances. While intrinsic dimension ($d$) and smoothness ($\beta$) are data properties, quantifying their impact on the convergence rate provides critical guidance for feature engineering and data preprocessing, and are central to real-world ML engineering. Our derived convergence rate is $O_p(|\Omega|^{-\frac{2\beta}{2\beta + d_{ui}}})$, where $d_{ui} = \text{max}(d_u, d_i)$. This theoretical insight offers specific actionable guidelines,
>
> - The dependence on $d_{ui}$ (intrinsic dimension) rather than $D$ (nominal dimension) theoretically justifies why two-tower models handle high-dimensional sparse inputs (like ID embeddings or bag-of-words) efficiently. However, it also suggests that reducing the intrinsic dimension of inputs (e.g., via pre-trained dense embeddings or PCA on raw features) before feeding them into the towers will strictly improve sample efficiency.
> - The dependence on $\beta$ infers that engineering features to be smoother with respect to the target ratings (e.g., using continuous semantic embeddings rather than discrete identifiers) directly accelerates convergence.
> - The bound allows engineers to estimate the required data volume $|\Omega|$ to achieve a target error rate given the estimated complexity ($d$) of their user/item features, moving capacity planning from guesswork to a grounded estimation.
>
> **W2A**: We clarify that the primary contribution of this paper is theoretical filling a significant gap in the literature. As noted in the introduction, despite the two-tower model's ubiquity in industry, its statistical guarantees and asymptotic behaviors were previously unexplored. The experiments were designed not merely to show state-of-the-art performance (which is indeed known), but to corroborate the theoretical findings, specifically,
> - Verifying that the model's performance scales with sample size $|\Omega|$ according to our predicted rates.
> - Demonstrating the model's superior handling of auxiliary covariates compared to matrix factorization baselines (SVD++), validating our theoretical formulation that $R(x_u, \tilde{x}_i)$ is a function of covariates, not just latent IDs. The novelty lies in bridging the gap between empirical success and mathematical understanding, ensuring that practitioners know why the model works and when it might fail (e.g., high intrinsic dimension).

---

> > ### Author Response · Authors · 2025-11-20
> > **Response to Reviewer 6buT (2/2)**
> >
> > **Questions:**
> >
> > **Q1A**: We did not vary the nominal dimension ($D$) in the reported plots because our theory suggests it has a negligible effect on performance compared to the intrinsic dimension ($d$). According to the manifold hypothesis utilized in our proofs (Section 4), the two-tower model learns to approximate the function on the low-dimensional manifold of the data. As long as the network width is sufficient to encompass the intrinsic manifold, increasing the ambient (nominal) dimension $D$ (assuming the intrinsic structure remains the same) does not degrade the convergence rate, which depends on $d$. This effectively highlights the model's robustness to the curse of dimensionality in the nominal space, a key theoretical advantage we wished to highlight.
> >
> > **Q2A**: The relationship is governed by the convergence rate bound derived in Theorem 4.5, $\|\hat{R} - R^*\|^2 \lesssim |\Omega|^{-\frac{2\beta}{2\beta + d_{ui}}}$. In our synthetic experiments, we fixed the number of observed ratings $|\Omega|$ (100k) while varying the matrix size $(n, m)$.
> > - Increasing the matrix size $(n, m)$ while keeping $|\Omega|$ fixed effectively increases the sparsity of the user/item space coverage. While our bound primarily depends on $|\Omega|$, the constant factors in the error bound (related to the covering number of the manifold) scale with the complexity of the space.
> > - When the intrinsic dimension is low, the model effectively learns the preference manifold even with sparse data. However, as the intrinsic dimension increases, the exponent $\frac{2\beta}{2\beta + d_{ui}}$ decreases. This makes the convergence significantly slower. Consequently, for higher intrinsic dimensions, the fixed sample size $|\Omega| = 100k$ becomes insufficient to cover the manifold, leading to a sharper degradation in performance as the problem space (matrix size) grows.
> >
> > **Q3A**: Our theory provides the mathematical justification for the cold-start capability. Intuitively, cold-start relies on covariates. However, without theory, it is not guaranteed that a model can effectively generalize from seen users ($x_{train}$) to unseen users ($x_{test}$) simply because they share a covariate space. Our consistency theorems prove that the estimated preference function $\hat{R}(x_u, \tilde{x}_i)$ converges to the true preference $R^*(x_u, \tilde{x}_i)$ over the domain of the covariates.
> > - Because the convergence is defined over the covariate manifold (Hölder space $\mathcal{H}(\beta)$), the error bound applies to any point on this manifold, including those corresponding to new users (cold-start) provided their features follow the same intrinsic distribution.
> > - The bound quantifies how much data is needed to solve the cold-start problem. It shows that warm users contribute to learning the mapping $f(x_u)$ for cold users with an efficiency determined by the smoothness $\beta$.
> >
> > **Q4A**: Thank you for pointing this out. The shading in the experiment tables is to highlight the best-performing model for each experimental configuration. In the revised manuscript, we will add these above clarifications as suggested.
> >
> > Please let us know if our response addresses your concerns.

---

### Meta-Review · Area_Chair_6LG5 · 2026-01-06

**Summary:**

The reviewers have several key concerns.

- The practical applicability of the theoretical guarantees is limited, as the identified factors (intrinsic dimension, smoothness) are data properties not easily controlled by practitioners.
-  The significance and relevance of the work to the ICLR community is also limited, given its focus on a specific industry model.
-  There is a mismatch between the claimed "ranking" guarantees and the actual metrics used (Top-k accuracy vs. position-sensitive ranking).

**Reviewer Concerns:**

The authors' rebuttal has addressed some of the reviewers' concerns. However, outstanding concerns remain unsolved, regarding limited practical guidance brought by the theoretical finding and limited core relevance and novelty for the ICLR audience, as the domain is still seen as niche, and the precise distinction between "retrieval" and "ranking" terminology, which the authors acknowledged but the fundamental framing of the work's contribution to broader machine learning theory was not decisively established.

**Reviewer Scores:**

Reviewer 6bUT (Score: 4): might keep the score unchanged or raise the score to weak accept, as the rebuttal clarified the experimental intent.

Reviewer rH1A (Score: 2): would likley keep the score (reject), as the fundamental concerns about the novelty and the narrow scope of the contribution are still major concerns.

Reviewer mrTr (Score: 4): would likely increase the score to weak accept, as the concerns were addressed.

---

### Decision · Program_Chairs · 2026-01-26

Reject